



# HORAYZON v1.1: An efficient and flexible ray-tracing algorithm to compute horizon and sky view factor

Christian R. Steger[1], Benjamin Steger[2], and Christoph Schär[1]

[1]Institute for Atmospheric and Climate Sciences, ETH Zürich, Zürich, Switzerland
[2]ESRI Research & Development Center Zürich, Zürich, Switzerland

**Correspondence:** Christian R. Steger (christian.steger@env.ethz.ch)

**Abstract.** Terrain parameters like topographic horizon and sky view factor (SVF) are used in numerous fields and applications. In atmospheric and climate modelling, such parameters are utilized to parameterise the effect of terrain geometry on radiation exchanges between the surface and the atmosphere. Ideally, these parameters are derived from a high-resolution digital elevation model (DEM), because inferring them from coarser elevation data induces a smoothing effect. Computing topographic

horizon with conventional algorithms is however slow, because large amounts of non-local terrain data have to be processes. We propose a new and more efficient method, which is based on a high-performance ray tracing library. By applying terrain simplification to remote topography, this allows the application of the new algorithms also with very high-resolution ($< 5$ m) DEM data, which otherwise would induce an excessive memory footprint. The topographic horizon algorithm is accompanied by a SVF algorithm, which was verified to work accurately for all terrain - even very steep and complex one. We compare the

computational performance and accuracy of the new horizon algorithm with two reference methods from literature and illustrate its benefits. Finally, we illustrate how sub-grid SVF values can be efficiently computed with the newly derived horizon algorithm for a wide range of target grid resolutions (1 - 25 km).

## 1 Introduction

In mountains, radiation exchange between the surface and the atmosphere is substantially influenced by terrain geometry.

By knowing local slope angle and aspect, the effect of self-shading on direct incoming shortwave radiation can readily be considered. However, for all other topographic effects on radiation, like topographic shading, (multiple) reflection of shortwave radiation and the exchange of longwave emission between slopes, the geometry of non-local terrain must be considered. For radiation modelling, two parameters are particularly relevant and often applied, the horizon and the sky view factor (SVF). The first parameter is uniformly defined in the literature and indicates the boundary line between the terrain and the sky as seen

from a certain location. It can be used to account for topographic shading, i.e. assessing if direct incoming shortwave radiation is blocked by surrounding terrain. The second parameter can be inferred from the horizon but is ambiguously defined: Zakšek et al. (2011) specifies the SVF as the solid angle of the visible sky, which corresponds to the fraction of a hemisphere occupied by the sky. Dozier and Frew (1990) provide a different definition of the SVF, which specifies the fraction of sky radiance a





location receives under the assumption of isotropic sky radiation. The latter definition is often used to parameterise effects like
terrain reflection of shortwave radiation and exchange of longwave emission between slopes.

The terrain parameters horizon and SVF are applied in a wide range of disciplines and fields: In atmospheric and climate modelling, topographic shading is considered in certain models by computing the terrain horizons of all grid cells (Chow et al., 2006; Arthur et al., 2018). In addition, some models use the SVF to correct fluxes of longwave and/or diffuse shortwave radiation (Müller and Scherer, 2005; Senkova et al., 2007; Buzzi, 2008; Manners et al., 2012; Liou et al., 2013; Rontu et al., 2016;
Lee et al., 2019). Terrain parameters in these studies are either computed from the model's internal elevation representation or from a sub-grid digital elevation model (DEM). Topographic shading is also considered in various spatially distributed land-surface models with typically higher horizontal resolutions - like in hydrology (Zhang et al., 2018; Marsh et al., 2020) and glaciology (Arnold et al., 2006; Olson and Rupper, 2019; Olson et al., 2019). For urban areas and cities, the SVF is utilised to quantify radiation exchanges in street canyons and their contribution to the urban heat island effect (Dirksen et al., 2019;
Scarano and Mancini, 2017). Additionally, SVF and horizon can also be used to estimate solar resources in urban environments (Calcabrini et al., 2019). In satellite climatology, the horizon line and SVF are crucial quantities to model radiation in complex terrain (Dürr and Zelenka, 2009; Bosch et al., 2010; Ruiz-Arias et al., 2010). In geochronology, a similar concept to the SVF is applied - the so-called topographic shielding (Codilean, 2006; Codilean et al., 2018). This quantity is used to correct incoming cosmic radiation fluxes, which can provide information on exposure ages of bedrock and surface denudation
rates. Finally, horizon lines are also relevant for more technical applications, like determining the camera position of an image by horizon matching. This technique can be used to geolocalise photographs (Pritt, 2012; Saurer et al., 2016), improve the estimated azimuth angles of augmented reality devices (Nagy, 2020) and even to localise a Mars Rover (Chiodini et al., 2017).

An early concept of computing horizon and SVF is presented in Dozier et al. (1981) and Dozier and Frew (1990). They propose an algorithm in which the horizon line for a position is computed by dividing the azimuth in discrete sectors. For
each sector, the horizon is derived by computing elevation angles of all DEM grid cells that intersect the sector's centre line and taking the maximum angle. A speed-up of this algorithm is suggested in Dozier et al. (1981), but the concept is only applied to DEM data on a regular grid with a uniform spacing. Bosch et al. (2010) suggested another approach to speed up horizon calculation. They divide a sector in a near-distance (<5 km) and far-distance domain. For the former domain, all DEM information is process whereas only a part of the information (local maxima) is used for the later. The horizon is then estimated
by combining the near- and far-distance horizons. Finally, Pillot et al. (2016) presents a horizon algorithm that closely follows the initial, non-accelerated concept of Dozier et al. (1981). In contrast to many earlier studies, they do not assume a planar DEM grid and account for the curvature of the Earth's surface.

In many studies (Pillot et al., 2016; Zhang et al., 2018; Marsh et al., 2020), horizon algorithms are still based on the conventional concept, in which all terrain information along a finite centre line is scanned to find the highest elevation angle.
These algorithms are typically sufficiently performant for DEMs with coarse resolutions and/or small sizes. Processing of large, high-resolution ($\leq$ 30 m) DEM data, like NASADEM (NASA JPL, 2020), USGS 1/3 arc-second DEM (USGS, 2017a) and swissALTI3D (Swisstopo, 2018) is however very time consuming. We propose a faster horizon algorithm, which is versatile in its application and based upon a state-of-the-art high-performance ray tracing library (Wald et al., 2014; Embree) used in





3D computer graphics. Such libraries are highly optimised and undergo continuous development, which make them attractive
for our purpose. In this approach, terrain information is stored as a triangular mesh in a bounding volume hierarchy (BVH),
and only a fraction of terrain information has to be checked along a search line (or ray). The proposed horizon algorithm is
accompanied by a SVF algorithm, which ensures accurate results for all terrain - even very steep and complex one. Additionally,
we illustrate how sub-grid SVF values can be computed efficiently for a large range of target grid spacings (1 - 25 km), which
was a main motivation to develop the new algorithms.

This paper is structured as follows: Input DEM data, which is used to evaluate the algorithms and illustrate computed terrain
parameters, is described in Sect. 2. Implementation details of the horizon and SVF algorithms are subsequently presented in
Sect. 3. In Sect. 4, the new algorithm is evaluated in terms of computational performance and accuracy. Section 5 shows how
the algorithm can be used to compute sub-grid SVF and illustrates its application with very high-resolution DEM data. Overall
conclusions and outlooks are presented in Sect. 6.

## 2   Data

To evaluate and illustrate the proposed horizon and SVF algorithms, we use data from three different DEMs with horizontal
resolutions ranging from $\sim 30$ to 2 m:

- **NASADEM** (NASA JPL, 2020) offers a horizontal resolution of 1 arc-second ($\sim 30$ m) and a near-global coverage (56°
  S to 60° N). NASADEM is the result of reprocessing Shuttle Radar Topography Mission (SRTM) data and incorporating
additional elevation data primarily from the Ice, Cloud, and Land Elevation Satellite (ICESat). Remaining voids were
  mainly filled with ASTER-derived Global DEM (GDEM). NASADEM elevations are referenced to the WGS84 ellipsoid
  and provided as orthometric heights relative to the Earth Gravitational Model 1996 (EGM96; Lemoine et al. 1998; NGA).

- **USGS 1/3 arc-second DEM** (USGS, 2017a) is provided at a horizontal resolution of $\sim 10$ m. The data set provides
  void-free and seamless elevation over the conterminous United States, Hawaii, Puerto Rico, other territorial islands, and
in limited areas of Alaska. The elevation data are referenced to the North American Datum of 1983 and orthometric
  heights are relative to the North American Vertical Datum of 1988.

- **SwissALTI3D** (Swisstopo, 2018) is a DEM with a very high resolution of 2 m and covers the entire area of Switzerland.
  The model was compiled from various sources: Below 2000 m a.s.l, LIDAR data with a high accuracy (in all three
  dimensions) of $\pm 0.5$ m is applied. Above, stereo correlation data are used, which has a accuracy of $\pm 1.0$ m to $\pm 3.0$ m.
Some manual updates regarding individual points, breaklines and areas were also included, which feature accuracies in
  the range of $\pm 0.1$ m to $\pm 1.0$ m. The elevation data are referenced to the the Swiss coordinate system LV95 and the Swiss
  national levelling network LN02.





## 3 Horizon and Sky View Factor algorithms

### 3.1 Preprocessing of Digital Elevation Model data

To compute the horizon with our applied ray tracing library, elevation data must be available in a Cartesian coordinate system. Furthermore, auxiliary quantities like local Upward-, North- and East-direction must be known as well as terrain slope angle and aspect for the successive SVF calculation. Elevation data sets are typically provided on map projections (SwissALTI3D) or in geodetic coordinates ($\phi$: latitude, $\lambda$: longitude) and orthometric height ($h_o$ (NASADEM and USGS 1/3 arc-second). A multi-level coordinate transformation is thus required and the auxiliary quantities must be computed.

### 3.1.1 Selection of required Digital Elevation Model domain

In a first step, we determine the total size of the DEM tile required to compute the horizon for a inner rectangular domain. The inner domain has to be extended by a boundary zone width $b$ according to the applied search distance for the horizon. Typical values for $b$ used in this study range from 25 - 50 km. For DEM data on an equally spaced grid, like SwissALTI3D, the extension of the inner domain is straightforward. For DEM data on a geodetic coordinate grid, like NASADEM and USGS 1/3

arc-second DEM, the size of the total domain is computed by extending the inner domain, which is bounded by $\lambda_{min}$, $\lambda_{max}$, $\phi_{min}$ and $\phi_{max}$, with $\Delta\lambda_a$, $\Delta\phi_s$ and $\Delta\phi_n$, respectively (Fig. 1).

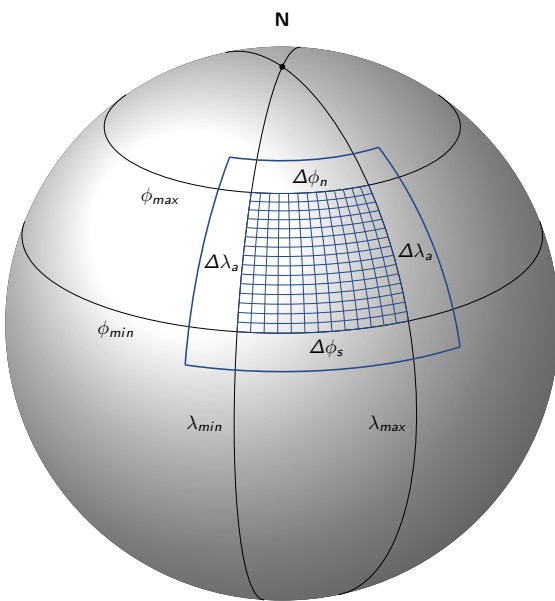

**Figure 1.** Total Digital Elevation Model domain (thick blue line) required to compute the horizon for the inner blue mesh. Black parallels of latitude represent circles and black meridians geodesics.





The required extension in the longitudinal direction can be approximated by

$$\Delta\lambda_a = \frac{2\pi}{p_l}\, b\,, \tag{1}$$

where $p_l$ represents the length of the parallel at $\phi_a = \max(\mid \phi_{min}\mid, \mid \phi_{max}\mid)$. This length is computed with

$$p_l = 2\pi\, \frac{a}{\sqrt{1 - e^2 \sin^2 \phi_a}}\cos\phi_a\,, \tag{2}$$

where $a$ represents Earth's equatorial radius (semi-major axis) and $e$ its eccentricity. The total required DEM input domain in the longitude direction is then given by subtracting, respectively adding, $\Delta\lambda_a$ from $\lambda_{min}$ and $\lambda_{max}$. The necessary extension in the latitudinal direction ($\Delta\phi_s$ and $\Delta\phi_n$) is computed from $b$ via the direct geodesic problem (Karney, 2013), whose equations are implemented in the C++ library GeographicLib and called via a Python wrapper (Karney). Computing domain extensions is more cumbersome in case geographic poles are included.

### 3.1.2 Coordinate transformations and computation of auxiliary quantities

We utilise multiple coordinate systems in this work, which are listed below and partially illustrated in Fig. 2:

– Map projection (optional)

– Geodetic coordinates

– Geocentric earth-centered, earth-fixed coordinates (ECEF)

– Global east-north-up coordinates (global ENU)

– Local east-north-up coordinates (local ENU)

– Spherical coordinates in the local ENU reference system

Definitions of these systems and transformations between them are provided in Appendix A. In case of DEM data on a map projection, we first transform the data to geodesic coordinates. For SwissALTI3D elevation data, we apply the equations provided in Swisstopo (2016). The following steps are then performed identically for all considered DEM products: First, we transform geodetic coordinates to a geocentric earth-centered, earth-fixed (ECEF) reference system $(x, y, z)$. Local Up, North and East directions are computed in this coordinate system for every DEM grid cell (see Appendix B1). DEM coordinates and direction vectors are then transformed to a topocentric east-north-up (ENU) reference system $(x', y', z')$. The origin of these coordinates coincides with the centre of the considered DEM domain and we refer to this system as global ENU coordinates. The transformation to a global ENU system constrains coordinates to a numerical range that can be represented with sufficient accuracy as single-precision floats. This data type is required in the applied ray tracing library. The above transformation steps are performed once for a certain DEM domain and the obtained DEM coordinates and direction vectors in global ENU coordinates are subsequently passed to the ray casting part of the algorithm. The size of the selected DEM domain is thereby primarily restricted by memory requirements.



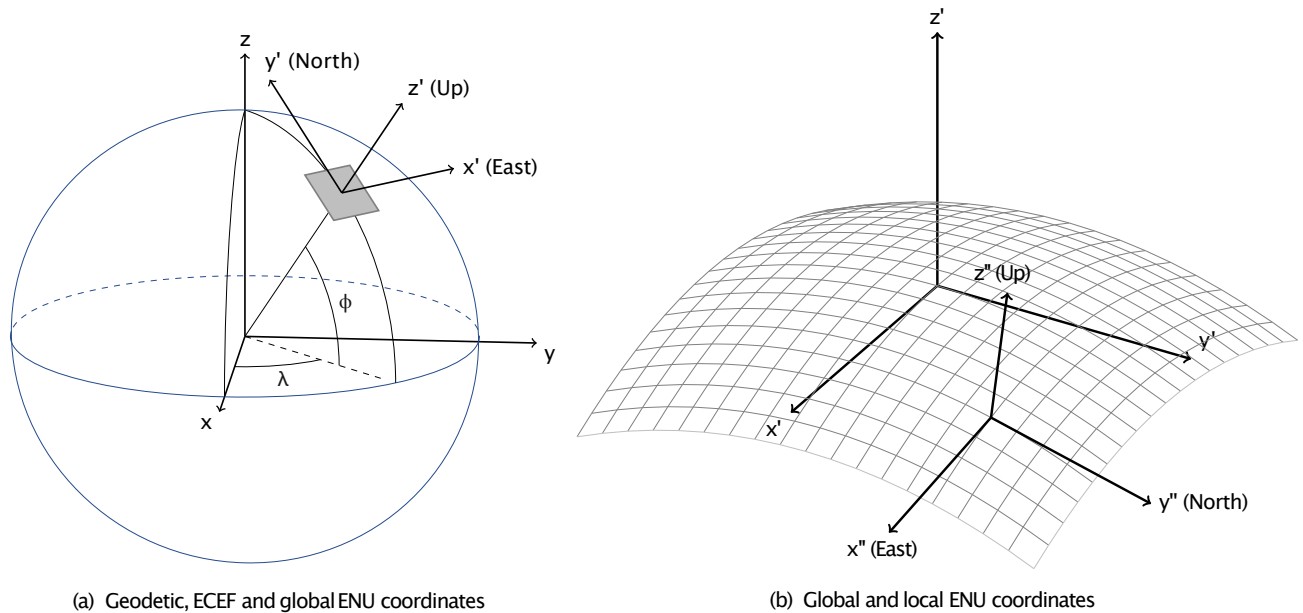

(a) Geodetic, ECEF and global ENU coordinates  (b) Global and local ENU coordinates

**Figure 2.** Illustration of the different coordinate systems applied: (a) the geodetic coordinate system ($\lambda$, $\phi$) and the geocentric earth-centered, earth-fixed (ECEF) reference system ($x,y,z$); (b) the topocentric east-north-up (ENU) reference system ($x',y',z'$), and a second local ENU system ($x'',y'',z''$). The grey mesh in panel (b) shows the curvature of the Earth's surface and illustrates how local and global ENU coordinates deviate.

Within the ray casting part of the algorithm (see Sect. 3.2.1), another topocentric reference frame is used - the so-called local ENU coordinate system ($x''$, $y''$ and $z''$). In this reference system, the z-axis is always aligned with the local upward direction (and thus the local ellipsoid normal; see Fig. 2b). The same reference system is applied to compute terrain slope aspect and angle (see Appendix B2). This ensures that Eq. (B6) can be solved for any topographic configuration (i.e. the matrix
is never singular). Finally, the SVF is computed in a spherical coordinate system, which is referenced to the local ENU system. All above steps regarding coordinate transformations and computations of auxiliary quantities were implemented in Cython (Behnel et al., 2011) and parallelised with OpenMP.

### 3.1.3   Masking of ocean grid cells

Computing the horizon from high-resolution elevation data is an expensive operation, even with the method presented in
this work. It is thus worthwhile to exclude areas, for which horizon information is either not needed or its computation is superfluous due to the non-existence of topography within a relevant radius. The latter applies to a large fraction of ocean grid cells. Unfortunately, such areas are not unambiguously masked in some DEM products. For instance, ocean grid cells in NASADEM have an elevation of 0 m - but inland areas might share the same value. We thus implemented a two-step method to address this issue: First, we label potentially relevant areas in the DEM product (for instance grid cells with an elevation of 0





m in NASADEM). Subsequently, we rasterise ocean coastlines from the Global Self-consistent, Hierarchical, High-resolution
Geography Database (GSHHG; Wessel and Smith 1996) to the same grid. Grid cells labelled as land in at least one of the two
data sets are classified as land and all remaining cells are treated as ocean. Coastlines are then retrieved from this raster as
contour lines. Finally, we have to find the shortest distance to the coastline for every ocean grid cell along a geodesic. This
procedure is expensive because an iterative algorithm is required to solve the inverse geodesic problem (Karney, 2013). We
therefore use the chord line (the closest straight line connecting two points on the geoid), which can readily be computed in
ECEF coordinates. To efficiently find the shortest distance, we transform coastline contours to ECEF coordinates and store them
in a SciPy k-d tree (Virtanen et al., 2020). A nearest neighbour query is then performed for ocean grid cells. Approximating
geodesics by chord lines is justifiable, because deviations between the two lines are small (e.g. $\sim$1 m for 100 km) for relevant
distances. Furthermore, chord lengths are always shorter than geodesics, which guarantees a conservative masking of ocean
grid cells.

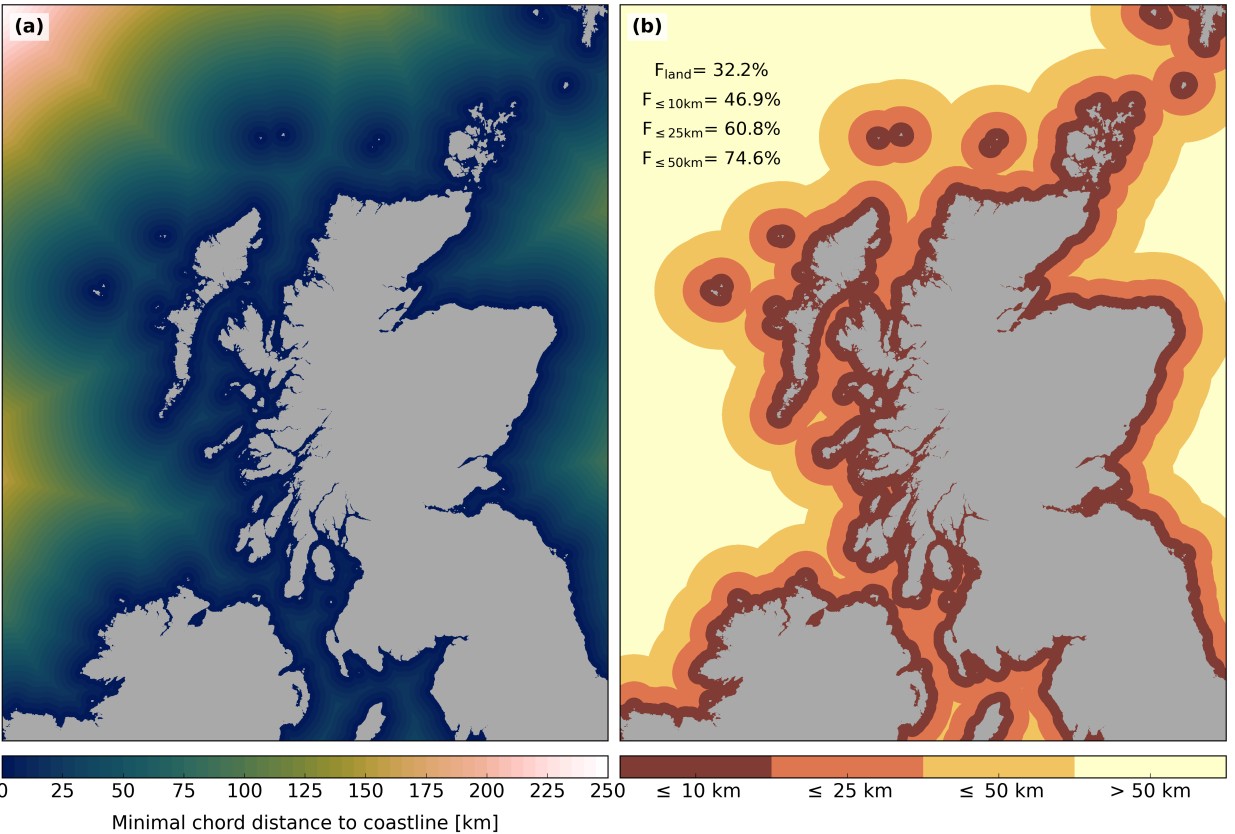

**Figure 3.** NASADEM grid ($21960 \times 32400$ cells) for the norther part of Great Britain and Ireland. (a) Chord distance for every grid cell to
the nearest coastline. (b) Ocean grid cells are categorized according to their minimal chord distance to the coastline in bands of $\leq 10$ km,
$\leq 25$ km and $\leq 50$ km. The numbers in the upper left indicate the fraction of remaining cells in case all ocean cells are masked ($F_{land}$) or
ocean cells exceeding a certain minimal distance to the coastline are masked.



Figure 3 shows the result of this masking approach for NASADEM data and a region covering the northern part of Great Britain and Ireland. By masking ocean grid cells entirely, horizon information has to be computed for only ∼32% of the total grid cells in the domain. This masking approach might e.g. be useful for land surface models. In case of considering ocean grid cells and applying a horizon search distance of 25 km, the masking allows to exclude ∼ 39% of all grid cells in the domain.

The implemented masking is not restricted to water grid cells and can be utilised to mask cells based on other criteria (e.g. land surface type, elevation or slope azimuth).

### 3.2 Computation of horizon by ray casting

#### 3.2.1 General implementation

We perform ray casting with the high-performance ray tracing library Intel Embree (Wald et al., 2014; Embree), which has been
released as open source under the Apache 2.0 License. In short, ray tracing in Embree works as follows: First, a BVH is built (in parallel) from input geometries, which can either be based upon *triangles*, *quads* (quadrilaterals) or a *grid* (quadrilaterals with a high tessellation level). This process recursively wraps the geometries in so-called bounding volumes, which form leaf nodes of a tree. The tree structure allows to perform the subsequent ray tracing in a highly optimised way: no children nodes of the tree have to be considered if a ray does not intersect with the parent node. In Cartesian coordinates, DEM data is typically
provided on a (almost) regular grid, thus all three Embree geometries can potentially be used. A performance test revealed that ray casting is fastest with *quads*, while using a *grid* allows for the fastest BVH building and reveals by far the smallest memory footprint.

In line with other algorithms (Dozier et al., 1981; Dozier and Frew, 1990; Pillot et al., 2016) , we compute the horizon for a location by splitting the azimuth angle in discrete sectors and sample along the centre lines. By default, we use 360 sectors.
Four different methods were tested to find the horizon within a sector with ray casting (Fig. 4). Because horizon detection results from discrete ray sampling, we have to define a desired accuracy for the horizon ($\alpha_r$), which we set to 0.25° by default. The simplest method, called *discrete sampling*, starts from a minimal elevation angle (-15.0° by default) and increments this angle until the ray no longer intersects terrain. The increment $\Delta\alpha$ is thereby set to 2 $\alpha_r$. The problem can be solved more efficiently by applying a *binary search* algorithm, which splits the elevation angle range sequentially. The desired accuracy
is reached as soon as the difference between the preceding and current ray elevation angle is smaller than 2 $\alpha_r$. Even faster methods can be obtained by considering the fact that the horizon represents a smooth continuous line. Horizon angles between two neighbouring sectors are thus typically very similar, particularly if a high number of azimuth sectors is used. We therefore implemented a third method, which estimates the horizon of the current sector from the previous one. The actual horizon is then found by applying discrete ray sampling from this angle. A fourth method was also tested, which estimates the horizon
of the current sector by linear extrapolation from the previous two sectors. However, this method did not result in a speed-up compared to the third method and was thus discarded. For all methods, the actual horizon angle lies within ±0.25° of the computed one. By assuming an uniform probability distribution of the actual horizon in the constrained range of the elevation angle (blue shaded area in Fig. 4), the mean error of the computed horizon is ±0.125°. To ensure that rays do not intersect

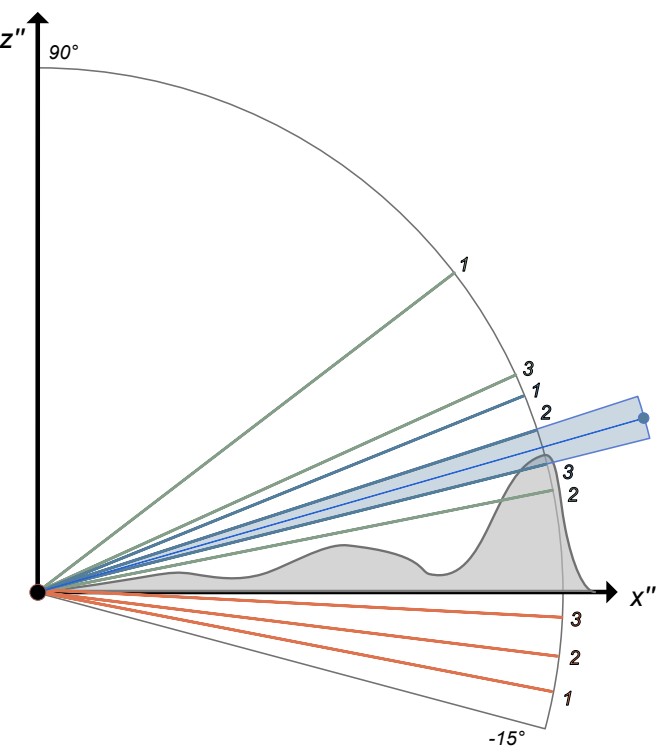

**Figure 4.** Overview of applied horizon detection algorithms with ray casting. Illustrated are the first three rays casted with the method *discrete sampling* (red), *binary search* (green) and *estimate from previous azimuth* (blue). In contrast to the text, an elevation angle spacing of $4°$ is used to allow for better readability. The blue dot marks the horizon estimate from the third method.

terrain directly at their origin due to numerical imprecision, the ray's origin is elevated by a small value of 0.01 m.

The ray casting part was implemented in C++ and parallelised with Intel Threading Building Blocks (TBB), which is recommended by Embree and also released under the Apache 2.0 License. In a first implementation, ray directions for a specific location were computed by rotating a vector, which initially points towards local North, in global ENU coordinates. This approach proved to be expensive due to the large number of trigonometric function evaluations. We accelerated this part by storing a discrete number of trigonometric functions, which are needed to compute all necessary ray direction in a local ENU

coordinate system. These vectors can subsequently be mapped to global ENU coordinates with Eq. (A5), which is considerably cheaper. Embree offers various options for building the BVH, which affects both BVH building time and subsequent ray casting. An evaluation of these options revealed that for our application, only the flags *robust* and *compact* have a significant impact on performance or the memory footprint of the algorithm. The implications of these flags are briefly addressed in Sect. 4.1 and 3.2.2, respectively.





### 3.2.2 Processing of elevation data with very high resolution

A disadvantage of the ray tracing based horizon algorithm is its larger memory demand compared to conventional horizon algorithms. Besides the DEM data, which requires 3 ($x$, $y$ and $z$ coordinate) $\times$ 4 Bytes per grid cell, additional information defining the connectivity of the triangle mesh and the BVH has to be stored. The memory requirements for this auxiliary data are smallest for the input geometry *grid* and were found to amount to an additional $\sim 90\%$ of the space the elevation data occupies. Applying the Embree flag *compact* did not lower these memory requirements any further (but revealed a significant impact on the memory footprint if *quads* were used). Currently, various DEMs with very high resolutions are available, like the USGS DEM $1$ m (USGS, 2017b), the ArcticDEM (2 m, Porter et al. 2018) and the swissALTI3D, which is available on resolutions of $0.5$ and $2$ m. In future, further products that cover larger geographical extents will likely become available. Processing such very high resolution DEM data can result in substantial memory requirements, as shown in the following example: The horizon of a $5 \times 5$ km domain should be computed from a $1$ m DEM with a horizon search distance of $25$ km. The elevation data alone requires $36.3$ GB of memory without considering space needed for building the BVH. These memory demands exceed the specifications of typical personal computers. However, memory requirements can be drastically reduced by simplifying terrain geometry in the outer boundary zone of the DEM domain (see Fig. 5). We perform this step with the heightmap meshing utility (hmm; Fogleman), which simplifies terrain based on a maximal allowed vertical error ($\Delta h$). Hmm is based on Garland and Heckbert (1995) and applies a greedy insertion algorithm and subsequent Delaunay triangulation to simplify terrain.

Figure 5 illustrates the setup in case terrain simplification is applied. An inner domain, which encompasses the area for which horizon values are computed, plus a boundary zone are represented by the full DEM information. The outer domain is split into for sub-domains and its terrain geometry is simplified to a triangulated irregular network (TIN). This step can be performed in parallel. Recombination of the five sub-domains introduces discontinuities in the triangulated surface, which are marked by red lines in Fig. 5. These discontinuities have to be patched - otherwise rays might pass through them without intersection terrain. We perform the patching by adding a vertical strip of triangles (also referred to as skirt; see Fig. 2 in Campos et al. (2020)) with a vertical extent of $3\Delta h$. As a consequence of the applied terrain simplification, the accuracy of computed horizon values decreases in case the horizon line is located in the outer DEM domain. The horizon accuracy due to terrain simplification $\alpha_s$ is thereby linked to the vertical error in terrain $\Delta h$ by

$$\alpha_s = 2 \arctan\left(\frac{\Delta h}{2\,d_m}\right), \tag{3}$$

where $d_m$ is the minimal distance between the area for which horizon values are computed and the simplified outer domain (see Fig. 5). This uncertainty in horizon accuracy adds to the one from the horizon detection algorithm ($\alpha_r$; see Sect. 3.2.1). Concretely, to e.g. meet a total total horizon accuracy of $0.25°$, one could apply the setting $\alpha_r = 0.15°$ and $\alpha_s = 0.1°$.





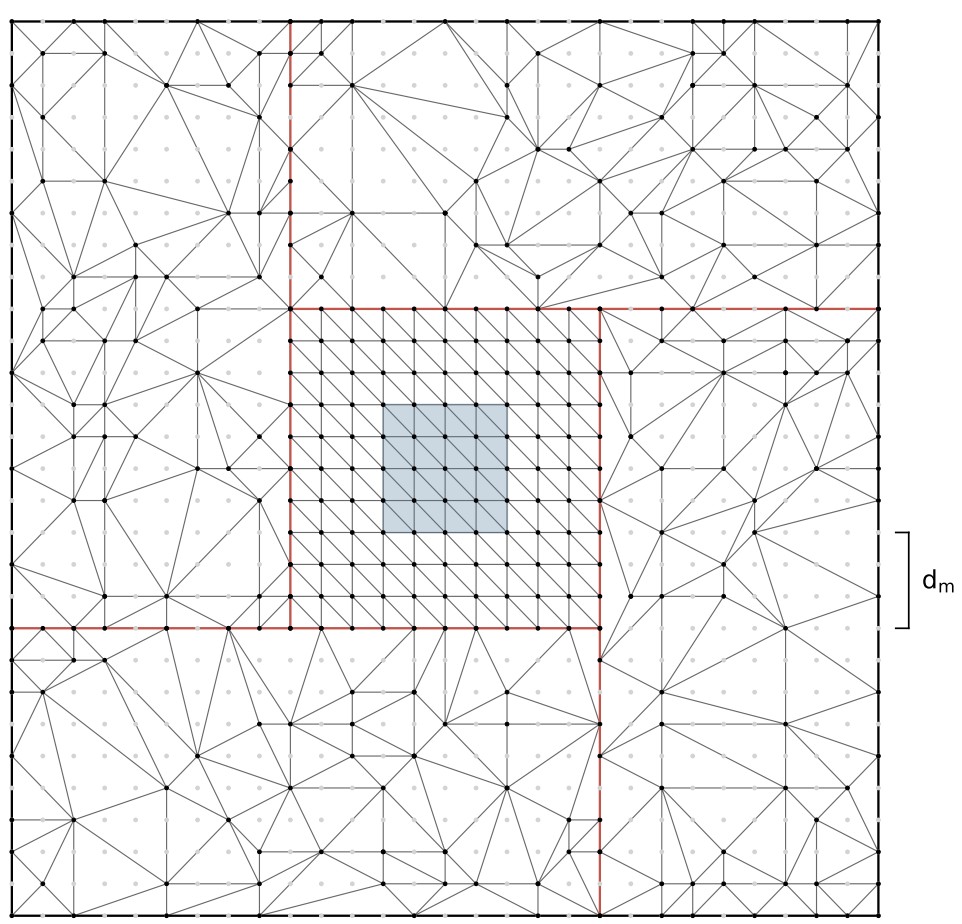

**Figure 5.** Illustration of triangulated surface in case of terrain simplification in the outer DEM boundary zone. The light gray dots represent grid cell centres of the original DEM data. The blue shaded domain shows the area for which horizon values are computed. Red lines mark discontinuities in the triangulated surface.





### 3.3 Sky View Factor computation

We consider the SVF definition, which yields the fraction of sky irradiance under the assumption of isotropic sky radiation, analogous to Dozier and Frew (1990) and Helbig et al. (2009). For an arbitrarily oriented surface, this SVF definition is expressed by

$$F_{sky} = \frac{1}{\pi} \int\limits_0^{2\pi} \int\limits_0^{\pi/2-\alpha_t} \cos\vartheta \sin\vartheta \, d\vartheta \, d\varphi \,, \tag{4}$$

with $\alpha_t$ representing the elevation angle of the surrounding terrain and $\vartheta$ and $\varphi$ the zenith and azimuth angle, respectively. Integration of Eq. (4) with respect to $\vartheta$ yields:

$$F_{sky} = \frac{1}{2\pi} \int\limits_0^{2\pi} \cos^2\alpha_t \, d\varphi \,. \tag{5}$$

This equation is identical to Eq. (8) in Helbig et al. (2009). An apparently straightforward way to compute $F_{sky}$ from the horizon, derived according to Sect. 3.2, is to transform horizon angles from the local ENU to a sloped coordinate system and use Eq. (5). However, this equation cannot be directly applied due to multiple reasons: First, it is no longer possible to represent horizon as a function of azimuth in the sloped coordinate system, because $\alpha_t$ can obtain multiple values for a certain $\varphi$ (see Fig. 6a). Secondly, if the surface normal intersects with surrounding terrain due to very steep terrain, then all horizon values are constrained to an azimuth range of $180°$ (Fig. 6b). And finally, the azimuth spacing of transformed horizon angles is no longer regular. Solving the double integral of Eq. (4) in the sloped coordinate system thus requires the consideration of more complex integration limits. Additionally, transformation of all horizon angles to a sloped coordinate system is expensive due to the evaluation of numerous trigonometric functions. We therefore compute the SVF in the horizontal local ENU coordinate system based on a modification of Eq. (4).

First, we compute the intersection of the inclined surface plane with a unit sphere. This plane passes through the origin of the local ENU coordinate system, thus we can write its implicit plane equation as

$$n_{x''}\, x'' + n_{y''}\, y'' + n_{z''}\, z'' = 0 \,, \tag{6}$$

with $\boldsymbol{n} = (n_{x''}, n_{y''}, n_{z''})$ being the normal vector of the inclined plane in local ENU coordinates. Combining Eq. (6) and (A7) yields the elevation angle of the plane-sphere intersection $\alpha_p$ as a function of the azimuth angle:

$$\alpha_p = \arctan\left( -\frac{n_{x''}}{n_{z''}} \sin\varphi - \frac{n_{y''}}{n_{z''}} \cos\varphi \right). \tag{7}$$

For a non-horizontal surface, we can generalise Eq. (4) by replacing $cos\vartheta$, which represents Lambert's cosine law, with

$$\cos\gamma = \begin{pmatrix} n_{x''} \\ n_{y''} \\ n_{z''} \end{pmatrix} \cdot \begin{pmatrix} \sin\vartheta \sin\varphi \\ \sin\vartheta \cos\varphi \\ \cos\vartheta \end{pmatrix}. \tag{8}$$



**Figure 6.** Visualisation of the horizon for two locations in the Lauterbunnen Valley (Bernese Oberland, Switzerland). (a) and (b) show the horizon in local ENU coordinates (grey) and in a sloped coordinate system (blue and red, respectively), whose z-axis is aligned with the surface normal. (c) illustrates the surrounding terrain of the two locations in an oblique view of the Lauterbunnen Valley from North. The blue and red lines represent the surface normals of the two locations.





Combing the generalised form of Eq. (4) with Eq. (8) yields

$$F_{sky} = \frac{1}{\pi} \int\limits_{0}^{2\pi} \int\limits_{0}^{\pi/2-\alpha_m} \left( n_{x''} \sin\vartheta \sin\varphi + n_{y''} \sin\vartheta \cos\varphi + n_{z''} \cos\vartheta \right) \sin\vartheta \, d\vartheta \, d\varphi, \tag{9}$$

where $\alpha_m = \max(\alpha_t, \alpha_p)$. Integration of Eq. (9) with respect to $\vartheta$ yields

$$F_{sky} = \frac{1}{2\pi} \int\limits_{0}^{2\pi} \left( (n_{x''} \sin\varphi + n_{y''} \cos\varphi) \left( \frac{\pi}{2} - \alpha_m - \frac{\sin(2\alpha_m)}{2} \right) + n_{z''} \cos^2\alpha_m \right) d\varphi. \tag{10}$$

260 This represents the analytical formulation of the SVF. To apply this equation with computed terrain horizon angles, we have to discretise it to

$$F_{sky} \approx \frac{\Delta\varphi}{2\pi} \sum_{i=1}^{M} (n_{x''} \sin\varphi + n_{y''} \cos\varphi) \left( \frac{\pi}{2} - \alpha_m - \frac{\sin(2\alpha_m)}{2} \right) + n_{z''} \cos^2\alpha_m, \tag{11}$$

where $M$ represents the number of equally spaced azimuth directions for which the horizon was computed. In principle, one could improve the accuracy of the numerical integration by employing Simpson's rule, but we believe that the overall uncer-
265 tainty is not determined by the errors of this integration, but rather by the computational resolution of the horizon computation.

In the literature, SVF computation is performed with different methods, which are for instance based on formulations from Helbig et al. (2009), Manners et al. (2012) and Dozier and Frew (1990). Computing the SVF in a sloped coordinate system with the equation suggested in Helbig et al. (2009), which is identical to Eq. (5), requires careful consideration of the integration limits. As illustrated by the blue and red dots in Fig. 6a) and b) and previously discussed, these limits can be complicated
270 for steep and complex terrain. Integration can be performed with the Trapezoidal rule and by summing up obtained areas - analogous to the area computation of a two-dimensional polygon. Apart from negligible numerical deviations, we obtained the same results with this method compared to applying Eq. (11). By testing the method of Manners et al. (2012), we believe to have found an error in its derivation: In Eq. (13) of Manners et al. (2012), angles are added that are expressed in different coordinate systems. The resulting error is minor for small slope angles but more pronounced for steeper terrain. Finally, we
275 considered the method suggested by Dozier and Frew (1990). By applying multiple trigonometric identities and considering the different coordinate systems applied, we found it to be identical to our solution. Concluding, if horizon angles are available in a horizontal coordinate system, it seems most convenient to perform the SVF integration in the same reference system.

## 4 Computational performance and accuracy of algorithms

We consider two reference horizon algorithms to evaluate the computational performance and accuracy of our method: First,
280 the algorithm described in Pillot et al. (2016), which is available as a MATLAB implementation (Pillot, 2016). Secondly, the algorithm applied in Buzzi (2008), which was implemented in the pre-processing tool of the limited-area atmospheric model COSMO (Steppeler et al., 2003) in Fortran and parallelised with OpenMP. Both algorithms are based on the conventional concept of sampling all elevation data along a centre line of an azimuth sector. The algorithm by Buzzi (2008) was developed





to process elevation data on a rotated latitude/longitude grid centred at the coordinate system's origin. The Earth's shape is fur-
thermore assumed to be spherical. Due to these restrictions, we processed the input DEM data for this section as follows: First,
we assume that DEM heights are referenced to a spherical Earth and ignore differences between orthometric and ellipsoidal
heights. Secondly, we bilinearly remapped DEM data to rotated longitude/latitude coordinates, whose origin is located in the
centre of the selected DEM domains. We apply two elevation data sets in this section: NASADEM with a horizontal resolution
of 30 m and USGS 1/3 arc-second DEM, which has a higher horizontal resolution of 10 m. For all algorithms, we used the
default number of 360 azimuth sampling sectors. For the ray-casting based algorithm, the elevation angle accuracy was set to
±0.25° and terrain simplifications according to Sect. 3.2.2 were not applied.

## 4.1 Evaluation of the computational performance

In the literature, suggested search distances for the horizon typically range from ca. 20 km (Senkova et al., 2007) to 50 km (Dürr
and Zelenka, 2009). These distances should be defined according to the desired horizon accuracy and the complexity of the
regional terrain. In areas like the Himalayas, elevation differences within 50 km can be as high as 7000 m (without intermediate
terrain obstruction), which corresponds to a horizon angle of ∼8°. For such terrains, an even larger search distance than 50 km
might be necessary if a high horizon accuracy is required. We applied NASADEM data, centred at Kleine Scheidegg (Bernese
Oberland, Switzerland) and USGS 1/3 arc-second DEM data, centred at Denali (Alaska, USA), for the performance evaluation.
The performance analysis of the ray-tracing based method revealed a dependency on terrain complexity and the algorithm's
performance is higher for simpler terrain. We therefore considered two additional domains for this algorithm, which are located
north of the above mentioned domains and feature less complex, hilly terrain. The overall performance was then computed as
an average between the two domains with different terrain complexity. The performance dependency on terrain is however
minor and in the order of ±10% from the average. All performance experiments have been carried out on a workstation with
an Intel Core i5 Quad-Core processor (3.4 GHz) with 16 GB of memory.

Figure 7 shows results from the performance analysis for two different horizontal DEM resolutions and horizon search
distances. The algorithm of Pillot et al. (2016) is not considered in this analysis because it was designed for point location
applications (its run time is substantially larger than the other two considered algorithms). Figure 7a reveals that the run time of
the conventional algorithm (Buzzi, 2008) scales distinctively with horizon search distance. For the ray tracing based method,
this is only true for relatively small (ca. $< 10^5$ grid cells) terrain sizes. This diverging pattern is caused by the varying ratio
of time spent on BVH building and ray tracing. For small domains, the BVH building contributes significantly to the total
run time, whereas for larger domains, this contribution is negligible. For larger domains (ca. $> 10^5$ grid cells), run times for
the ray casting algorithm are almost independent of the horizon search distance. For the DEM with 10 m resolution (Fig. 7b),
the performance analysis looks overall very similar. However, run times between the conventional and the ray-tracing based
algorithm diverge even further: for processing $10^6$ grid cells with a search distance of 50 km, the new algorithm reveals a
speed-up factor of ∼72 for a 30 m DEM, whereas this factor increases to ∼321 for a 10 m DEM. As mentioned in Sect.
3.2.1, Embree offers an option for robust BVH building, which we used for the analysis shown in Fig. 7. Disabling this option
increases ray tracing performance by approximately 20%. However, as a trade-off, the requirements for horizon accuracy are





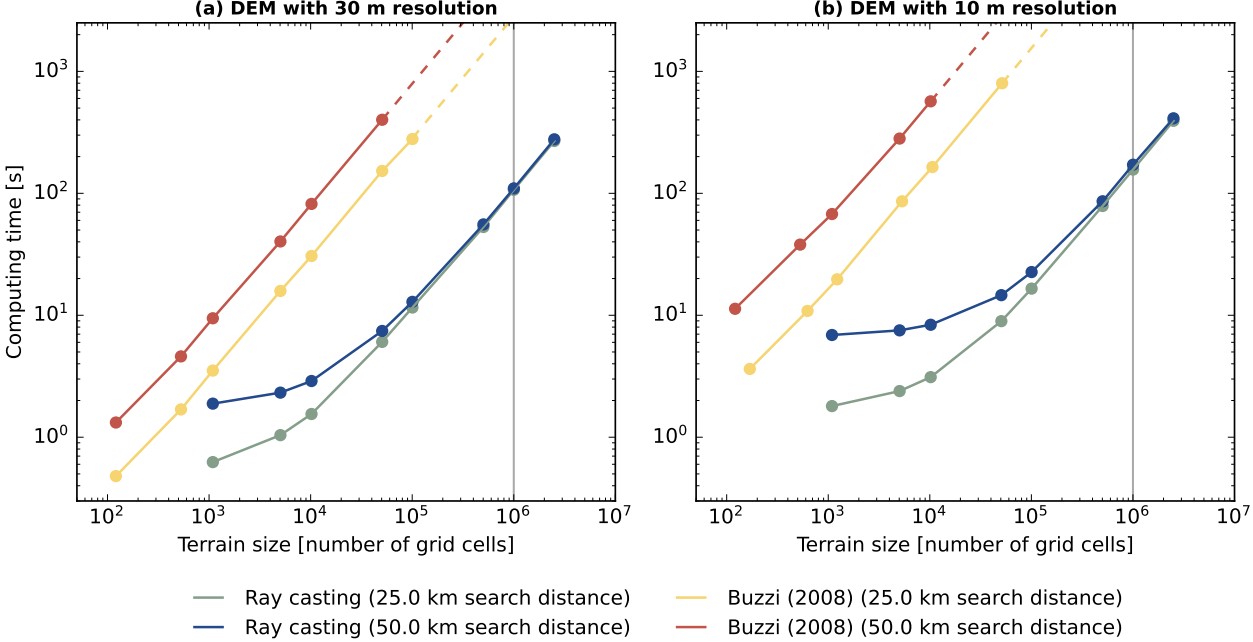

**Figure 7.** Computational performance of the ray casting algorithm relative to the one implemented by Buzzi (2008) in a log-log-plot. Panels (a) and (b) show the performance for a DEM with 30 and 10 m horizontal resolution, respectively. Dashed lines indicate a linear extrapolation of the performance relationship. Experiments were carried out on a Intel Core i5 Quad-Core processor (3.4 GHz) workstation with 16 GB of memory.

not longer strictly met because triangles from the terrain mesh might be missed by rays. We thus always enable the flag *robust* for BVH building, even if such errors were found to occur extremely infrequent.

In summary, the performance analysis revealed that the ray-casting method is much faster for all considered terrain sizes (by about two orders of magnitude). The speed-up increases with domain size, higher spatial DEM resolution and larger horizon search distances. The higher performance of the ray-tracing based algorithm is mainly caused by the more efficient storage of DEM data, which drastically reduces the elevation information that has to be considered along a sampling line. The relevance of this effect grows both with increased DEM resolution and horizon search distance.

## 4.2 Accuracy evaluation for real terrain

We compared the accuracy of the ray tracing based horizon algorithm to the methods suggested by Buzzi (2008) and Pillot et al. (2016). The accuracy of the latter algorithm was assessed by in-situ horizon measurements collected with a Theodolite in Corsica (Pillot et al., 2016). We evaluated the different algorithms for an approximately 10 by 10 km wide NASADEM domain ($324 \times 324$ grid cells) centred at Kleine Scheidegg (Bernese Oberland, Switzerland). To increase overall accuracy of the computed horizon lines, we enhanced the search distance for the horizon to 100 km. Due to the comparably high run time





of the Pillot et al. (2016) algorithm, we did not apply it to the full domain. Instead, we ran it for 1000 cells within the $324 \times 324$ domain, which were drawn by random uniform sampling.

**Figure 8.** Horizons for three locations computed with the three different algorithms. The boxes at the lower boundary of the panels illustrate the distance to the horizon line based on the ray casting algorithm. The numbers in the panel's upper left show the geographic longitude ($\lambda$) an latitude ($\phi$) of the location. The numbers at the right boundaries of the panels represent the mean ($\Delta_{mean}$) and maximal ($\Delta_{max}$) spread in computed horizon angles for the location. The three methodologies considered qualitatively agree, but note the occurrence of staircase-like behaviour for the Pillot and Buzzi algorithms.

Figure 8 shows obtained horizon lines for three example locations. In Fig. 8a, the distance to locations forming the horizon is generally larger than 1 km and the agreement between the three algorithms is, with a mean spread of $0.42°$, very good. For

locations shown in 8b and 8c, the agreement between the algorithms deteriorates and maximal deviations up to $14.34°$ occur. Considering the horizon distance information, it is obvious that the inferior agreement is constrained to azimuth angles with





close distances to the horizon. For these ranges, both reference algorithms indicate staircase-shaped changes in the horizon line, which is e.g. also apparent in Fig. 9 in Pillot et al. (2016). Table 1 indicates that findings from the three example locations translates to the entity of analysed locations. The agreement between the three algorithms is considerably smaller for locations

and azimuths with close ($< 1$ km) proximity to the horizon line – both in terms of statistical mean and $95^{th}$ percentile. Additionally, Table 1 reveals that the agreement between the ray casting and Buzzi (2008) algorithm is consistently better than between the other two combinations. Deviations from the algorithm by Pillot et al. (2016) are higher for large horizon distances. A potential explanation for this pattern might be the way how sampling for a certain azimuth direction is implemented in Pillot et al. (2016), which happens along a loxodrome.

**Table 1.** Absolute differences in horizon angles [°] between the three algorithms. The mean and the $95^{th}$ percentile (p95) are shown for all data and grouped according to the associated distance to the horizon.

| Comparison \ Horizon distance | all | | < 1 km | | ≥ 1 km | |
|---|---|---|---|---|---|---|
| | mean | p95 | mean | p95 | mean | p95 |
| Ray casting – Pillot et al. (2016): | 1.51 | 7.78 | 2.54 | 10.92 | 0.82 | 3.14 |
| Ray casting – Buzzi (2008): | 1.00 | 4.25 | 1.80 | 7.10 | 0.48 | 0.82 |
| Pillot et al. (2016) – Buzzi (2008): | 1.33 | 5.50 | 1.96 | 9.08 | 0.91 | 1.94 |

The pronounced staircase-shaped artefacts in Fig. 8b and 8c, which cause the poor agreement between the algorithms for close horizon distances in Table 1, are induced by the non-smooth terrain representation in the reference algorithms of Buzzi (2008); Pillot et al. (2016). These algorithms assume uniform elevations within grid cells with vertical drops at the cells' edges. The non-smooth terrain representation introduces two disadvantages: First, the occurrence of unnatural steps in the horizon line (see Fig. 8c) and secondly, a high sensitivity of the computed horizon on the chosen azimuth angle. The relevance of

these issues increases with decreasing distance between the centre location and the horizon. If computed horizon lines are used for SVF calculations, the issue of artificial steps is partially attenuated because terrain horizon $\alpha_t$ is occasionally exceeded by plane horizon $\alpha_p$ and thus not considered (see Sect. 3.3). However, a part of the error does propagate to computed SVF values. In the ray casting algorithm, gridded DEM data are converted to a triangle mesh, which represents a smooth surface. Subsequently, this algorithm does not suffer from the aforementioned issues.

**4.3 Verification with idealised terrain geometries**

To quantitatively verify our methodology, we additionally assessed the implemented horizon and SVF algorithms by means of two idealised z-axis-symmetric terrain geometries. The first one, called *Crater*, represents a simple hemispherical cavity, which was also considered in Manners et al. (2012). Its elevation $h$ is defined according to

$$h(d) = \begin{cases} r - \sqrt{r^2 - d^2} & \text{if } d < r, \\ r & \text{if } d \geq r, \end{cases} \qquad (12)$$





where $d$ is the distance from the centre and $r$ the radius of the cavity. Except for the rim of the cavity, the terrain represented by this geometry is concave, which implies that horizon lines are almost exclusively formed by non-adjacent terrain. To cover the other case, we consider a second, partially convex geometry (*Crater hill*), whose elevation is defined by

$$h(d) = \begin{cases} 0.5\, r\, f_a \left( \cos\left(\frac{2\,d\,\pi}{r}\right) + 1.0 \right) & \text{if } d < \frac{r}{2} \\ r - 2\sqrt{r\,d - d^2} & \text{if } \frac{r}{2} \le d < r \;, \\ r & \text{if } d \ge r \end{cases} \tag{13}$$

where $f_a$ represents the amplitude factor, which determines the height of the central bump. Cross-sections of the two geometries
are shown in Fig. 9a for the parameter setting r = 1000 m and $f_a$ = 0.9.

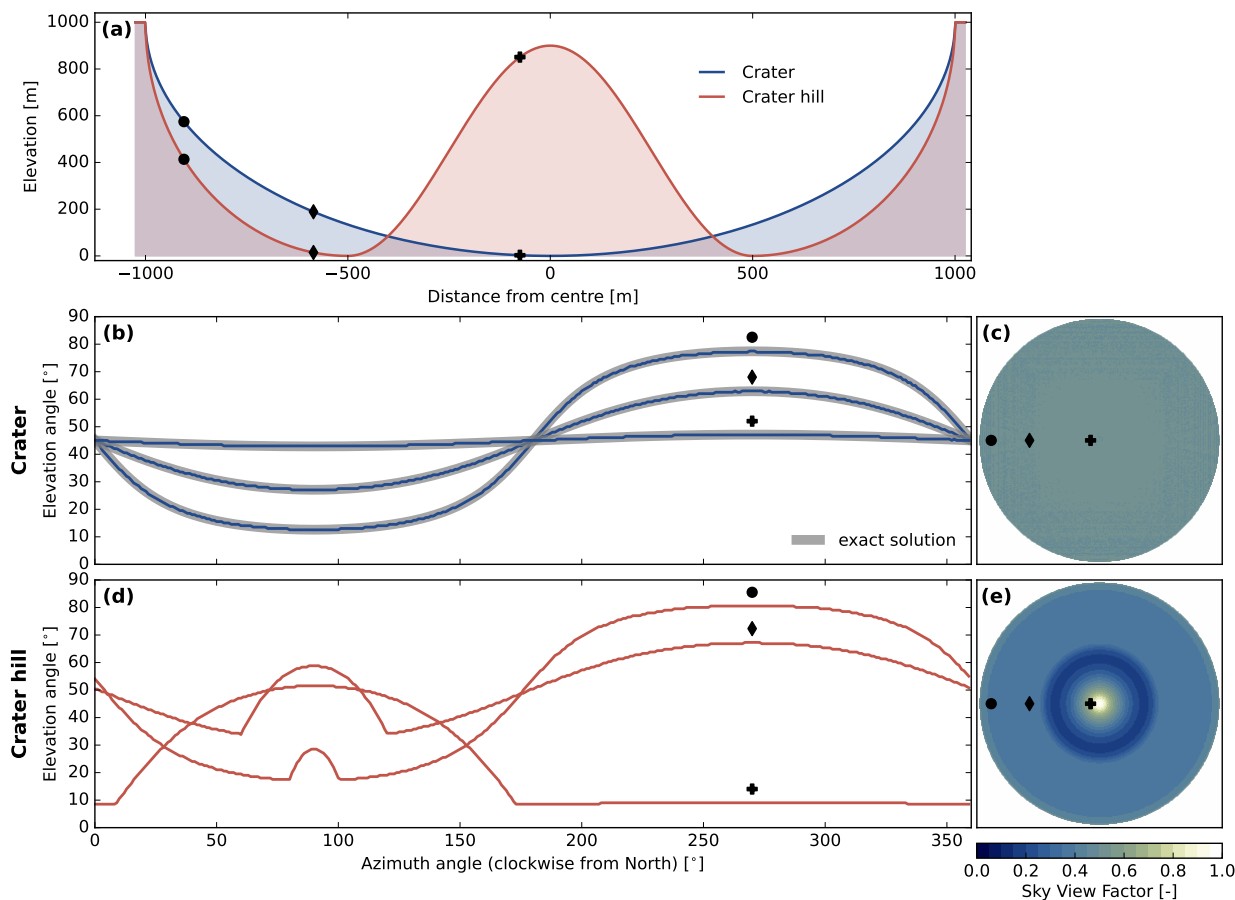

**Figure 9.** Terrain parameters for the two idealised geometries *Crater* (blue) and *Crater hill* (red). (a) Elevation cross-sections of the two terrains, where black symbols mark locations for which the horizon is illustrated in (b) and (d). In (b), exact solutions (grey lines) for the horizon are shown in addition to the ones computed with ray casting. (c) and (e) illustrate horizontally resolved SVF for the *Crater* and *Crater hill* geometries, where the black symbols indicate again the locations considered in (b) and (d).





We discretised both terrains by $1026 \times 1026$ grid cells and computed the horizon with the default setting of 360 azimuth sectors and an accuracy of $0.25°$. The resulting horizon for three grid cells, whose location is marked in Fig. 9a, is illustrated in panel (b) (*Crater*) and (d) (*Crater hill*) of the same figure. In accordance with the smooth surfaces of the geometries, horizon angles represent smooth lines without artefacts. In case of the *Crater* geometry, the horizon is invariably formed by the rim of

the cavity and its exact solution can thus readily be derived. Figure 9b indicates that horizon lines computed with ray casting align with these 'perfect' solutions. Obtained spatial SVF values for both geometries are shown in Fig. 9c and 9e. The SVF for the *Crater* geometry is uniformly 0.5 within the cavity (with negligible numerical deviations), which is in line with the analytical solution derived in Manners et al. (2012). The SVF for the *Crater hill* geometry is spatially variable with lowest values around $d = r/2$ and highest values in the centre.

It is possible to validate the resulting SVF values of both geometries, at least in a horizontally aggregated way, by physical and geometrical considerations: Manners et al. (2012) illustrates in Sect. 2.4 that the same horizontally aggregated long-wave flux is emitted from a flat disc and a hemispherical cavity. This relation holds for any cavity - not only the perfectly hemispherical one. We can rearrange Eq. (24) and (25) of Manners et al. (2012) to

$$F_{sky,disc} = \frac{A_{cavity}}{A_{disc}} F_{sky,cavity} \,, \tag{14}$$

where $A_{cavity}$ and $A_{disc}$ represent the surface area of the cavity and the flat disc, and $F_{sky,disc}$ the SVF of the disc, which is exactly 1. Applying Eq. (14) to the *Crater* and *Crater hill* geometries yields $\sim$0.999 for both cases, which confirms the correct implementation of the horizon and SVF algorithm.

## 5 Application examples of the algorithms

In this section, we present example applications of the ray tracing based horizon and SVF algorithm. Output from Sect. 5.1 and
5.2 can be used to parameterise the effect of terrain on surface radiation in weather and climate models. Sect. 5.3 illustrates the computation of horizon and SVF from very high-resolution elevation data. These outputs are primarily interesting for very high-resolution land-surface models applied in mountainous terrains.

### 5.1 Computation of terrain parameters and sub-grid sky view factor

As mentioned in the introduction, terrain parameters like horizon and SVF are already applied in several numerical weather
and climate models to account for topographic effects on radiation. In some models, terrain parameters are derived from the model's internal elevation representation, which typically features a grid spacings of $> 500$ m to $\sim 100$ km. This elevation data is normally smoothed to ensure numerical stability of the model. The relative coarse spacing (and the potential smoothing of orography) lead to a smoothing of terrain parameters - i.e. computed horizons angles are typically lower and obtained SVF values higher. In other models, terrain parameters are computed from a sub-grid scale DEM and subsequently spatially
aggregated. In the latter case, DEM data with high spatial resolution has to be processes, which can be done efficiently with



**Figure 10.** Terrain parameters computed from NASADEM for a 40 km wide window centred at Lauterbunnen Valley (Bernese Oberland, Switzerland). Panels (a) to (c) illustrate the elevation, slope angle and slope aspect. Note that aspects with slopes $< 1°$ are masked. Panel (d) shows the computed SVF on the native resolution of $\sim 30$ m and panels (e) and (f) the spatial aggregation of this parameter to 3 and 12 km, respectively.

our ray tracing based algorithm. Such a sub-grid scale parameterisation is for instance presented in Helbig and Löwe (2012), which emulates the effects of terrain reflection of shortwave radiation on surface albedo.

We illustrate the computation of terrain parameters and the spatial aggregation of the SVF by means of two DEMs with different resolution, the NASADEM and the USGS 1/3 arc-second DEM. Output from the NASADEM is shown in Fig. 10 for
a 40 km wide domain centred at Lauterbunnen Valley (Bernese Oberland, Switzerland). The horizon was computed with the default setting of 360 azimuth sectors and a search distance for the horizon of 50 km. Panels (d) to (f) illustrate how the range of





**Figure 11.** Terrain parameters computed from USGS 1/3 arc-second DEM for a 40 km wide window centred at Denali (Alaska, USA). Panels (a) to (c) illustrate the elevation, slope angle and slope aspect. Note that aspects with slopes < 1° are masked. Panel (d) shows the computed SVF on the native resolution of ∼ 10 m and panels (e) and (f) the spatial aggregation of this parameter to 3 and 12 km, respectively.




SVF changes with spatial aggregation. 3 and 12 km are common horizontal resolutions applied in regional climate modelling (Ban et al., 2021; Sørland et al., 2021; Jacob et al., 2014). Fig. 10f shows that, even on a relatively coarse resolution of 12 km, the aggregated SVF of some grid cells is still significantly smaller than 1.0. Fig. 11 illustrates terrain parameters computed from the USGS 1/3 arc-second DEM for a 40 km wide area centred at Denali (Alaska, USA). The geographic latitude of this area is relatively high ($\sim 63°$ North), which means that topographic shading is a very relevant process due to low solar elevation angles – particularly during Northern Hemispheric winter. In contrast to NASADEM processing, the number of azimuth sectors was decreased to 60 to reduce the required storage space for the 3-dimensional horizon information. Compared to Fig. 10, obtained SVF values on the native grid are generally lower. This translates to the spatially aggregated SVF values and even on a scale of 12 km, almost the entire area features SVF values below $0.85$ (see Fig. 11f).

## 5.2 Accelerated computation of sub-grid Sky View Factor

The application of sub-grid SVF in weather and climate models, as for instance in Hao et al. (2021), requires the computation of high-resolution horizon for large domains spanning several hundred to thousand kilometres. This step is computationally expensive, even with the new horizon algorithm (with the default setting of $N_a = 360$ and $\alpha_r = 0.25°$) presented in this study. We thus tested two approaches to decrease computational time further while maintaining a high level of accuracy: On the one hand, we decreased the number of azimuth sectors and the horizon accuracy.

**Table 2.** Different geographic domains used to test the accelerated computation of sub-grid SVF values.

| Label | Geographical Location | Latitude/ longitude[a] [°] | Domain size [km] | Spacing [m] | Mean slope[b] [°] | Mean SVF[b] [-] |
|---|---|---|---|---|---|---|
| Central Alps | Switzerland/ Austria/Italy | 46.663/10.393 | 100 x 100 | 30 | 26.5 | 0.87 |
| Grand Canyon | USA | 36.130/-111.970 | 50 x 50 | 30 | 16.3 | 0.91 |
| Hawaii | USA | 22.050/-159.540 | 50 x 50 | 30 | 8.6 | 0.97 |
| Kamchatka | Russia | 55.920/160.500 | 50 x 50 | 30 | 11.9 | 0.97 |
| Karakoram | Pakistan/ China | 35.883/76.513 | 100 x 100 | 30 | 28.7 | 0.83 |
| Fiordland | New Zealand | -44.750/168.100 | 50 x 50 | 30 | 29.0 | 0.82 |
| Patagonia | Argentina/ Chile | -49.271/-73.043 | 100 x 100 | 30 | 12.3 | 0.95 |
| Three Parallel Rivers | China | 28.186/98.871 | 100 x 100 | 30 | 31.8 | 0.83 |
| Yosemite | USA | 37.750/-119.600 | 50 x 50 | 30 | 16.7 | 0.94 |
| Denali | USA | 63.069/-151.008 | 50 x 50 | 10 | 24.8 | 0.85 |

[a]Geographic coordinates refer to the centre of the domain.

[b]Computed with the reference sampling density ($\sim$1024 samples per $km^2$).



These two parameters are interdependent in the fastest horizon detection method (see Sect. 3.2.1). If e.g. only $N_a$ is decreased, the average number of rays applied per sampling location and sector increases due to larger horizon differences between neighbouring sectors. We thus only considered settings in which both $N_a$ and $\alpha_r$ are altered simultaneously to keep

the averaged number of rays per grid cell and sector similar. We considered three combinations beside the default setting: $N_a$ = 60 & $\alpha_r$ = 1.5°, $N_a$ = 30 & $\alpha_r$ = 3.0° and $N_a$ = 15 & $\alpha_r$ = 6.0°, which represent performance increase factors of ~6, ~12 and ~24. On the other hand, we reduced spatial sampling density. In our reference setting, we computed the horizon ~1024 times per $km^2$, which approximately aligns with the spatial resolution (~30 m) of NASADEM. We performed decreases in spatial sampling density by considering subsets of the reference sampling locations, which were drawn randomly and spatially

uniform. To obtain more robust results, we repeated the random drawing $10^4$ times. We performed these tests for 10 different geographic domains, which cover a broad range of geomorphologies (see Table 2). DEM data for this analysis was bilinearly remapped to grid spacings of ~31.3 m (NASADEM) and ~10.4 m (USGS 1/3arc-second DEM). In these remapped products, and area of ~1 $km^2$ contains $32 \times 32$ or $96 \times 96$ DEM cells, respectively. Spatial SVF aggregation scales of 1, 3, 12 and 24 km are considered, which represent common horizontal resolutions for numerical weather and climate models.

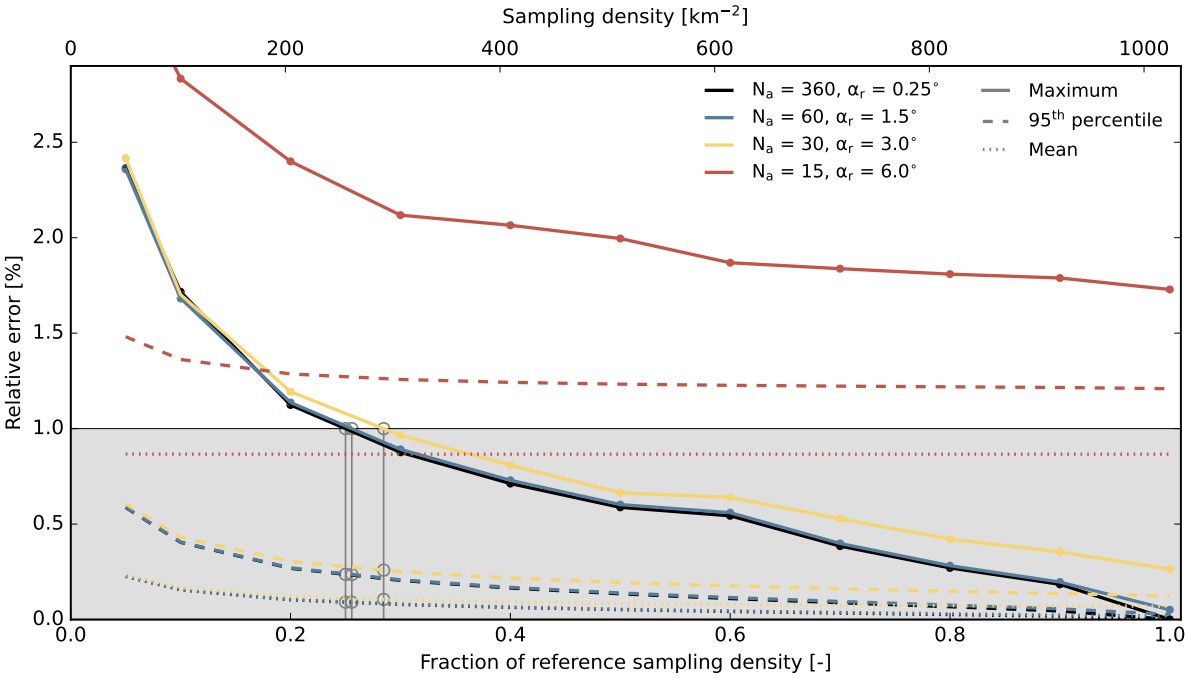

**Figure 12.** Example of the relative error in spatially aggregated SVF for the Central Alps region as a function of sampling density. The figure displays errors from $10^4$ randomly and spatially uniformly drawn samples for an aggregation to 3 km. $N_a$ denotes the number of applied azimuth sectors and $\alpha_r$ the horizon accuracy. The lower x-axis shows the sampling density relative to the reference density (~ 1024 samples per $km^2$) while the upper x-axis indicates absolute values. Intersections of the coloured solid lines, which represent the maximal relative error, with the horizontal 1% error line are denoted by gray circles and vertical lines.





Figure 12 illustrates the obtained results for the Central Alps region and a spatial aggregation of the sub-grid SVF values to 3 km. As expected, decreasing sampling density induces a gradual increase in relative SVF error in all considered error statistics (maximum, $95^{th}$ percentile and mean). Remarkably, this gradual behaviour is less evident for the applied sets of $N_a$ and $\alpha_r$. Initial decreases of $N_a$, until $N_a = 30$, revealed minor effects on the accuracy of the computed sub-grid SVF. Only the decrease from $N_a = 30$ to $N_a = 15$ showed a clear deterioration in the accuracy of sub-grid SVF. The same behaviour was also found

for other geographical regions. The grey circles and vertical lines in Fig. 12 illustrate the required sampling density to meet a maximal relative error of 1% and the associated $95^{th}$ percentile and mean of this error.

**Figure 13.** Overview of error statistics for the applied speed-up tests of sub-grid SVF calculation. The upper row illustrates the required spatial sampling density to meet a maximum relative error of 1% in sub-grid SVF computation. The lower row shows the associated $95^{th}$ percentiles and means in relative error. Settings ($N_a$ and $\alpha_r$) for which the maximum relative error never falls below 1% are hatched. Note the variable y-axis range between panels in the upper row.





Figure 13 displays these values for all considered regions and spatial aggregation scales. Values for different regions are typically clustered in relatively narrow bands. Apparently, the Fiordland region in New Zealand exhibits the highest terrain complexity, as it typically requires the highest sampling density to meet the 1% maximum relative error (see upper row in
Fig. 13). In contrast, the Kamchatka region in Russia requires a comparably low sampling density. This can be explained by the relative simple terrain geometry of this region, which is shaped by Stratovolcanos. For all conducted experiments, the $95^{th}$ percentile and the mean in relative error remain below 0.5% and 0.2% - except for the experiment with $N_a = 15$ (see lower row in Fig. 13). Regarding speed-up factors for computing sub-grid SVF with a maximal absolute error of 1%, the following conclusions can be drawn from Fig. 13: For a spatial aggregation to 1 and 3 km, the setting $N_a = 30$ & $\alpha_r = 3.0°$
is most favourable, while a decrease in the spatial sampling density is not (or only to a minor extent) possible. This allows a performance increase, relative to the default setting, of a factor of $\sim$12. For the spatial aggregation to coarser resolutions (12 and 25 km), the setting $N_a = 30$ & $\alpha_r = 3.0°$ is again optimal. For these resolutions, the sampling density can additionally be reduced to 5% (12 km) and 2.5% (25 km) of the reference density. This yields total speed-up factors of $\sim 240$ ($12 \times 1.0/0.05$) and $\sim 480$ ($12 \times 1.0/0.025$) for sub-grid SVF computations for resolutions of 12.0 and 25.0 km, respectively.

An earlier method to compute sub-grid scale SVF was presented in Helbig and Löwe (2014). They developed a model, which estimates spatially aggregated SVF from local terrain parameters, which are cheap to compute. This model is faster than our approach but also exhibits larger relative errors in computed SVF - particularly for target grids with high spatial resolutions (1.0 - 2.5 km). The choice of model, i.e. Helbig and Löwe (2014) versus our approach, depends on the available computational resources and the desired accuracy for the sub-grid scale SVF. An advantage of our approach is its potential to seamless derive
accurate sub-grid SVF values for all spatial scales.

## 5.3   Application to very high resolution DEM data

In this section, we demonstrate the application of the horizon and SVF algorithm with very high resolution DEM data. As mentioned in Sect. 2, we used SwissALTI3D data with a horizontal resolution of 2 m. To lower the memory footprint of the high resolution data during processing, we simplified terrain representation in the boundary zone of DEM according to Sect.
3.2.2. We computed terrain parameters for two $3 \times 3$ km domains in the Glarus Alps in Switzerland.

The first domain is centred around Tödi and is overall convex-shaped (Fig. 14), while the latter is centred at Limmerensee and features a rather concave-shape terrain geometry (Fig. 15). For the absolute horizon accuracy, we selected a value of 0.25°, which is partitioned to $\alpha_r = 0.15°$ and $\alpha_s = 0.1°$. For $d_m$ (see Fig. 5), we chose a distance of 7 km. For the domain centred around Tödi, a search distance for the horizon of 30 km was applied. Choosing a larger distance is not possible due to the
limited spatial coverage of SwissALTI3D data. Without terrain simplification, the memory footprint of the DEM data amounts to $\sim 11.9$ GB, while with terrain simplification, total memory requirements can be lowered to $\sim 0.92$ GB ($\sim 867$ MB for the inner domain and $\sim 55$ MB for the outer TIN). For this domain, obtained SVF values are highest and close to 1.0 in the centre, which features a high-elevated glaciated plateau. The lowest SVF values, which frequently fall below 0.4, typically coincide with steep walls that are e.g. found north and south-west of the plateau. Further areas with very low SVF values can be found
in the southern-eastern region and are caused by glacier crevasses.

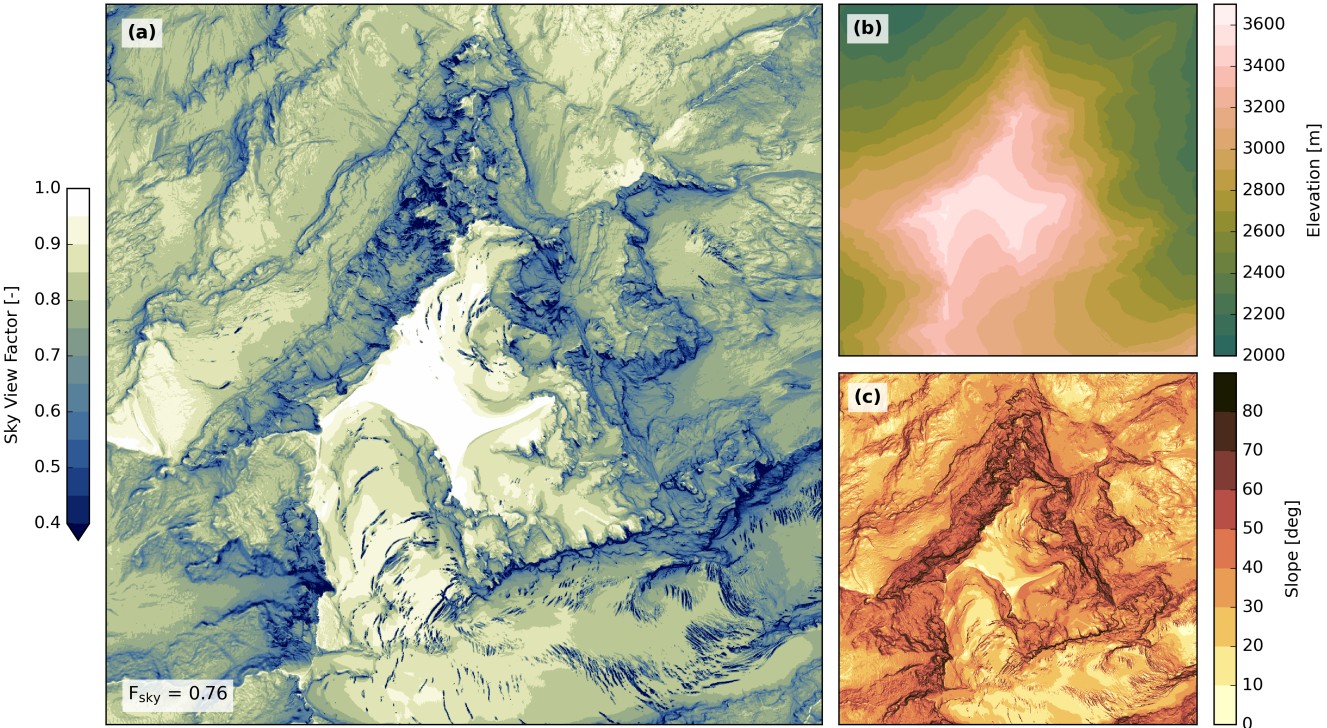

**Figure 14.** Terrain parameters computed from 2 m SwissALTI3D DEM for a 3 km wide domain centred at Tödi (Glarus Alps, Switzerland). Panel (a) illustrates the spatially gridded SVF, while the number in the lower left represents the domain-wide average. Panels (b) and (c) show the associated surface elevation and slope angles.

For the second considered domain (Fig. 15), the horizon search distance could be enhanced to 35 km. By considering the full DEM information, memory demands for the DEM data would have amounted to $\sim 16.0$ GB - without considering memory needed to build the BVH. With terrain simplification, these demands dropped to $\sim 0.94$ GB ($\sim 867$ MB for the inner domain and $\sim 73$ MB for the outer TIN). In contrast to Fig. 14a, the spatially aggregated SVF is lower and averages to 0.71. This

relatively low value is primarily caused by the deep gorge in the north-western part of the domain, which features a larger coherent area with SVF values below 0.5.

## 6 Conclusions

Horizon and derived SVF are used in various fields and applications. Conventional horizon algorithms typically process the full elevation information along an azimuth sector's centre line, which makes them slow for (very) high resolution DEMs. We

propose an new and more efficient method, which is based on a high-performance ray-tracing library. In this approach, terrain information is stored in a tree structure (BVH) and only a fraction of elevation data have to be considered along a scanning line. A comparison of the ray-tracing based horizon algorithm with a conventional method revealed its high computational





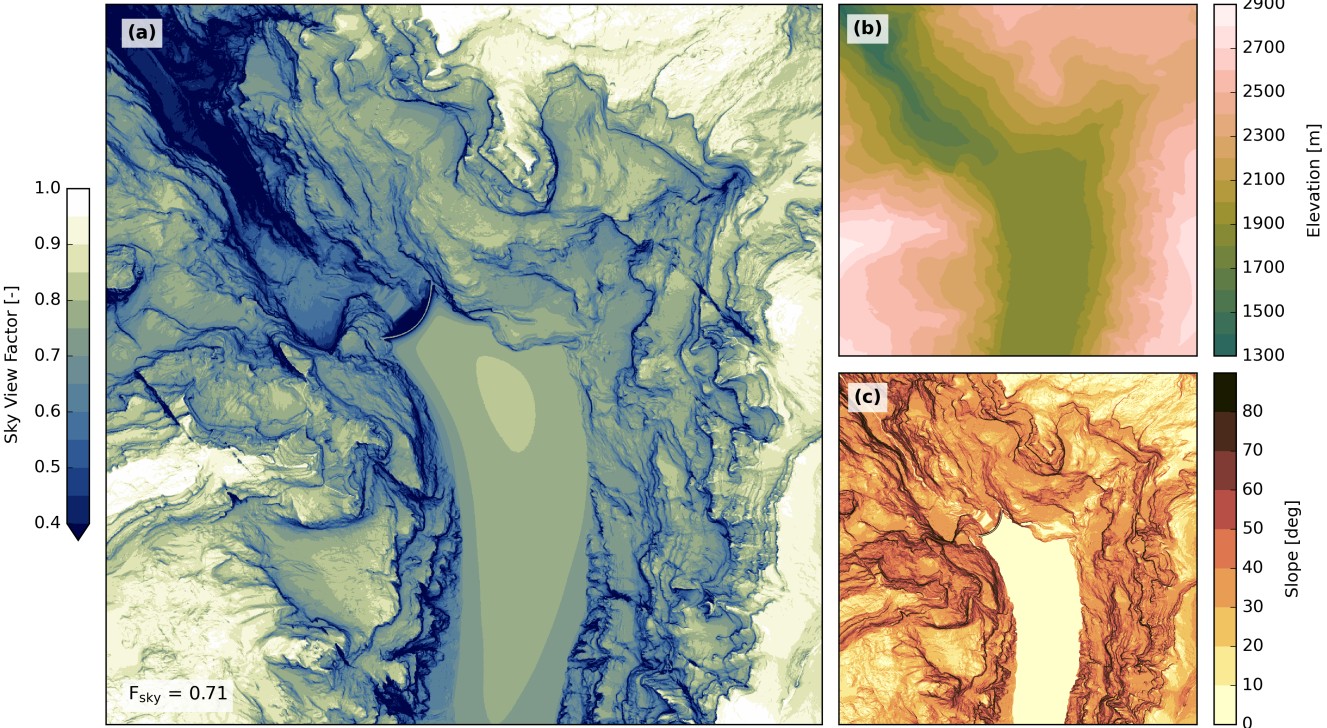

**Figure 15.** Terrain parameters computed from 2 m SwissALTI3D DEM for a 3 km wide domain centred at Limmerensee (Glarus Alps, Switzerland). Panel (a) illustrates the spatially gridded SVF, while the number in the lower left represents the domain-wide average. Panels (b) and (c) show the associated surface elevation and slope angles.

performance, which amplifies for higher DEM resolutions and larger horizon search distances. The new algorithm exhibits only a minor performance dependency on horizon search distance, which allows to compute more accurate horizon angles by

considering larger search distances. Applying the horizon (and SVF) algorithm to larger domains can additionally be accelerated by masking water grid cells, whose minimal distance to the coastline is larger than the search distance for the horizon. In terms of accuracy, the ray tracing based algorithms agrees well with two existing methods in case of larger distances (> 1 km) between the sampling location and the horizon line. For smaller distances, deviations are larger. These discrepancies are caused by differences in internal terrain rendering. In the two reference algorithms, terrain is represented by quadrilaterals

with a uniform elevation within individual cells, which results in a staircase-shaped surface. These structures translate to the computed horizon lines. In the new algorithm, terrain information is rendered by a triangle mesh, which represents a smooth continuous surface. Computed horizon lines have subsequently more natural gradients and do not suffer from staircase-shaped artefacts induces by terrain representation. A disadvantage of the new algorithm is its larger memory footprint, which is critical if very high resolution DEM data is processed. However, these memory demands can be drastically lowered by simplifying

terrain in the outer boundary zone of the DEM domain.





To infer SVF values from computed horizon angles, various methods are suggested in the literature (Dozier and Frew, 1990; Helbig et al., 2009; Manners et al., 2012), which are either applied in a horizontal or sloped coordinate system. We tested these methods for very steep and complex terrain and concluded that, in case horizon angles are available in a horizontal coordinate system, the method by Dozier and Frew (1990) is most convenient to apply. This SVF algorithm yields correct results for all

terrains - even very steep and complex ones.

The terrain parameters horizon and/or Sky View Factor are applied in various numerical weather and climate models (Müller and Scherer, 2005; Chow et al., 2006; Senkova et al., 2007; Buzzi, 2008; Manners et al., 2012; Liou et al., 2013; Rontu et al., 2016; Arthur et al., 2018; Lee et al., 2019) to parameterise the effects of terrain geometry on surface radiation - either on the scale of the model grid or on a sub-grid scale. The relevance of the SVF for parameterising the effect of topography

on surface radiation was confirmed in a recent study by Chu et al. (2021). They showed that, on domain-averaged scales, results from a three-dimensional ray-tracing simulation agree well with a SVF-based parameterisation. Even with our efficient horizon algorithm, the computation of sub-grid SVF is expensive for large weather and climate model domains. However, we demonstrated that the computational time can be further reduced by considering less accurate horizon lines and by reducing the spatial sampling density. The loss in SVF accuracy is thereby only minor. The speed-up factor growths with increasing

resolution differences between the DEM and the target grid and exceeds 400 for the coarsest considered target resolution (25 km). The proposed method to efficiently computed sub-grid SVF provides a complement to the statistical method suggested by Helbig and Löwe (2014), which links SVF to local terrain parameters and is thus computationally very cheap. The choice of method depends on the available computational resources and the desired accuracy in SVF.

A run time analysis of our horizon algorithm revealed a considerable speed-up compared to conventional algorithms. How-

ever, performance could likely be further improved with the following suggestions, which concern the performance-critical ray tracing part: We currently do not apply coherent rays in Embree, which allow for a more efficient utilisation of the BVH. Considering such ray streams might speed up horizon detection. To increase run times for workstations with performant graphic processing units (GPUs), ray casting could be performed with a GPU-based ray tracer, like NVIDIA OptiX (Parker et al., 2010).

*Code and data availability.* HORAYZON is made available under the terms and conditions of the MIT license. The source code has been archived on Zenodo (https://doi.org/10.5281/zenodo.6369224) and is also available from GitHub (https://github.com/ChristianSteger/HORAYZON). HORAZON depends on Intel Embree, Intel Threading Building Blocks (TBB) and the NetCDF4 C++ library. A Python installation with the packages NumPy, Cython, SciPy, pyproj, GDAL, Shapely, scikit-image, GeographicLib, Fiona and PyGEOS is furthermore required. HORAYZON is a cross platform application (Windows, Linux and macOS) and supports both x86 and ARM architectures. On multi-core

processor systems, HORAYZON can be run in parallel via TBB. All DEM data used in this study is freely available from the respective source stated in the reference.





## Appendix A: Coordinate transformations

Various Cartesian and spherical/elliptical coordinate systems are used in this work. In terms of Cartesian coordinates, we apply three different systems: ECEF $(x, y, z)$, global ENU $(x', y', z')$ and local ENU $(x'', y'', z'')$ coordinates. Equations to transform
between the different reference systems are provided below.

### A1  Transformation from Geodetic to ECEF coordinates

Transformation from Geodetic to ECEF coordinates is performed by

$$
\begin{aligned}
x &= (N(\phi) + h_e)\cos\phi \cos\lambda \\
y &= (N(\phi) + h_e)\cos\phi \sin\lambda \\
z &= \left(\frac{b^2}{a^2}N(\phi) + h_e\right)\sin\phi
\end{aligned}
\tag{A1}
$$

with $N(\phi) = a/\sqrt{1 - e^2 \sin^2\phi}$ and $e^2 = 1 - b^2/a^2$. $\phi$ represents the geodetic latitude, $\lambda$ longitude, $a$ the equatorial Earth
radius (semi-major axis), $b$ the polar Earth radius (semi-minor axis) and $e$ the eccentricity.

### A2  Transformation from ECEF to global ENU coordinates

Transformation from ECEF to global ENU coordinates is achieved by

$$
\begin{pmatrix} x' \\ y' \\ z' \end{pmatrix} =
\begin{pmatrix}
-\sin\lambda_r & -\cos\lambda_r & 0 \\
-\sin\phi_r \cos\lambda_r & -\sin\phi_r \sin\lambda_r & \cos\phi_r \\
\cos\phi_r \cos\lambda_r & \cos\phi_r \sin\lambda_r & \sin\phi_r
\end{pmatrix}
\begin{pmatrix} x - x_r \\ y - y_r \\ z - z_r \end{pmatrix},
\tag{A2}
$$

with $\phi_r$ and $\lambda_r$ representing the geodetic latitude and longitude. $x_r$, $y_r$ and $z_r$ constitute the coordinates of the tangential point
in ECEF coordinate. The transformation of a vector $\boldsymbol{b}$ from ECEF to global ENU coordinates is performed with

$$
\begin{pmatrix} b_{x'} \\ b_{y'} \\ b_{z'} \end{pmatrix} =
\begin{pmatrix}
-\sin\lambda_r & -\cos\lambda_r & 0 \\
-\sin\phi_r \cos\lambda_r & -\sin\phi_r \sin\lambda_r & \cos\phi_r \\
\cos\phi_r \cos\lambda_r & \cos\phi_r \sin\lambda_r & \sin\phi_r
\end{pmatrix}
\begin{pmatrix} b_x \\ b_y \\ b_z \end{pmatrix}.
\tag{A3}
$$

### A3  Transformation between global and local ENU coordinates

We distinguish between two topocentric reference systems, the global and local ENU coordinates. The axes of the global ENU coordinate system do not coincide with local East, North and upward directions of all DEM grid cells. This is only true for
local ENU coordinates. A vector $\boldsymbol{b}$, expressed in global ENU coordinates ($x'$, $y'$ and $z'$), is converted to local ENU coordinates ($x''$, $y''$ and $z''$) by

$$
\begin{pmatrix} b_{x''} \\ b_{y''} \\ b_{z''} \end{pmatrix} = R
\begin{pmatrix} b_{x'} \\ b_{y'} \\ b_{z'} \end{pmatrix},
\tag{A4}
$$





where $R$ represents the rotation matrix. The inverse transformation requires the inverse of the rotation matrix $R^{-1}$. Rotation matrices represent orthogonal matrices, thus their inverse is identical to their transpose. The inverse transformation can subsequently be written as:

$$
\begin{pmatrix} b_{x'} \\ b_{y'} \\ b_{z'} \end{pmatrix} = R^T \begin{pmatrix} b_{x''} \\ b_{y''} \\ b_{z''} \end{pmatrix}.
\tag{A5}
$$

The rotation matrix $R$ is defined by

$$
R = \begin{pmatrix} a_{e,\,x'} & a_{e,\,y'} & a_{e,\,z'} \\ a_{n,\,x'} & a_{n,\,y'} & a_{n,\,z'} \\ a_{u,\,x'} & a_{u,\,y'} & a_{u,\,z'} \end{pmatrix},
\tag{A6}
$$

where $\boldsymbol{a}_e$, $\boldsymbol{a}_n$ and $\boldsymbol{a}_u$ represent the local ENU coordinate axes East, North and Up in global ENU coordinates.

## A4 Transformation between local ENU and spherical coordinates

In the local ENU coordinate system, we also apply spherical coordinates

$$
\begin{aligned}
x'' &= \cos\alpha\,\sin\varphi \\
y'' &= \cos\alpha\,\cos\varphi\;. \\
z'' &= \sin\alpha
\end{aligned}
\tag{A7}
$$

In this reference system, the azimuth angle $\varphi$ is measured clockwise from North ($y''$) and the elevation angle $\alpha$ is measured from the $x'' - y''$-plane. In terms of the zenith angle $\vartheta$, which is measured from Up ($z''$), the transformation can be expressed as

$$
\begin{aligned}
x'' &= \sin\vartheta\,\sin\varphi \\
y'' &= \sin\vartheta\,\cos\varphi\;. \\
z'' &= \cos\vartheta
\end{aligned}
\tag{A8}
$$

The elevation and zenith angle are linked via the relation $\alpha = (\pi/2) - \vartheta$.

## A5 Conversion from orthometric to ellipsoidal height

Elevation data of DEMs often refers to orthometric heights. These elevations are measured relative to a geoid and must be converted to ellipsoidal heights, if coordinates are subsequently transformed from a geodetic to a geocentric coordinate system. The relation between the ellipsoidal height $h_e$, the orthometric height $h_o$ and the geoid undulation $N$ is specified by the following equation (Pillot et al., 2016; Grohmann, 2018):

$$
h_e = h_o + N.
\tag{A9}
$$





For NASADEM data, we computed the undulation $N$ based on EGM96 (Lemoine et al., 1998; NGA). For USGS 1/3 arc-second
570  elevation data, the GEOID12A geoid model (NGS) is applied. We bilinearly interpolate $N$ from a 5 arc-minutes (EGM96) or
1 arc-minute (GEOID12A) reference grid.

## Appendix B:  Computation of auxiliary quantities

### B1    Local East, North and Upward unit vectors

Computing the horizon line for a certain location requires knowledge about the local direction vectors pointing towards East,
North and Up. We compute these unit vectors in ECEF coordinates according to the following equations. The upward vector
is represented by the ellipsoid normal vector and can be computed as

$$\begin{pmatrix} a_{u,x} \\ a_{u,y} \\ a_{u,z} \end{pmatrix} = \begin{pmatrix} \cos\phi\cos\lambda \\ \cos\phi\sin\lambda \\ \sin\phi \end{pmatrix}. \tag{B1}$$

This vector ($\boldsymbol{a}_u$) is also called geodetic normal or $\boldsymbol{n}$-vector. The vector $\boldsymbol{a}_n$, pointing towards North and being perpendicular to
vector $\boldsymbol{a}_u$, can readily be derived in ECEF coordinates. First, the vector between the current location ($\boldsymbol{v}_l$) and the North Pole
$\boldsymbol{v}_p$ is computed as

$$\boldsymbol{v}_n = \boldsymbol{v}_p - \boldsymbol{v}_l. \tag{B2}$$

The North Pole is given by $\boldsymbol{v}_p = (0, 0, b)$. Vector $\boldsymbol{v}_n$ is then projected on the location's normal plane to receive

$$\boldsymbol{v}_j = \boldsymbol{v}_n - (\boldsymbol{v}_n \cdot \boldsymbol{a}_u)\,\boldsymbol{a}_u. \tag{B3}$$

Vector $\boldsymbol{a}_n$ is obtained by normalising vector $\boldsymbol{v}_j$

$$\boldsymbol{a}_n = \frac{\boldsymbol{v}_j}{\|\boldsymbol{v}_j\|}. \tag{B4}$$

The East unit vector ($\boldsymbol{a}_e$) is simply computed as

$$\boldsymbol{a}_e = \boldsymbol{a}_n \times \boldsymbol{a}_u. \tag{B5}$$

The direction vectors are subsequently transformed to global ENU coordinates with Eq. (A3).

### B2    Slope aspect and angle of terrain

To compute the SVF, local terrain slope an aspect must be known, which can be represented by a local surface tilt vector.
Various slope algorithms exist (Jones, 1998; Corripio, 2003) which typically consider the nearest four to eight DEM grid cells.
We select an approach, in which a plane is fitted to the respective centre grid cell and its eight neighbours. The plane fitting





is performed in the local ENU coordinate system by minimising the sum of squared $z''$-differences between the plane and the nine cells. This approach requires solving a linear system of equations defined as

$$
\begin{pmatrix}
\sum_{i=1}^{9} x_i''^2 & \sum_{i=1}^{9} x_i'' y_i'' & \sum_{i=1}^{9} x_i'' \\
\sum_{i=1}^{9} x_i'' y_i'' & \sum_{i=1}^{9} y_i''^2 & \sum_{i=1}^{9} y_i'' \\
\sum_{i=1}^{9} x_i'' & \sum_{i=1}^{9} y_i'' & 9
\end{pmatrix}
\begin{pmatrix}
n_{x''} \\
n_{y''} \\
n_{z''}
\end{pmatrix}
=
\begin{pmatrix}
\sum_{i=1}^{9} x_i'' z_i'' \\
\sum_{i=1}^{9} y_i'' z_i'' \\
\sum_{i=1}^{9} z_i''
\end{pmatrix},
\tag{B6}
$$

where $\boldsymbol{n}$ represents the surface normal of the sloped plane and $x''$, $y''$ and $z''$ the DEM coordinates of the nine grid cells. The same method is used in the Geographic Information System software ArcGIS (ArcGIS).

*Author contributions.* CRS, BS and CS designed the research framework. Code implementation and data analysis was conducted by CRS. CRS, BS and CS were involved in writing and revising of the manuscript.

*Competing interests.* The authors declare that there is no conflict of interest regarding the publication of this article.

*Acknowledgements.* Graphics were made using Python Matplotlib (version 3.4.3), PyVista (version 0.32.1), tikz (version 3.1.2) and Affinity Designer (version 1.10.4). Scientific colour maps of Crameri et al. (2020) are used.



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
