# Peer review of "HORAYZON v1.2: An efficient and flexible ray-tracing algorithm to compute horizon and sky view factor"

_Geoscientific Model Development, 2022_

## Referee Comment (RC2)

**Review of "HORAYZON v1.1: An efficient and flexible ray-tracing algorithm to compute horizon and sky view factor" by Steger et al**

**Main comments**

The paper introduces the model HORAYZON for computating horizon lines and sky view factor (SVF) from digital elevation models which is relevant for land surface modeling/NWP. The model introduces a new algorithm based on the ray-tracing library `embree` to cope with demanding computational requirements of SVF computations in high-res modelling. Standard computations are notoriously slow since the SVF is a non-local quantity of the DEM.

The manuscript is well written, contains illustrative, high-quality figures, benchmark tests, comparisons to previous algorithms (Buzzi) and the application to three large DEM data-sets (NASADEM, swisstopo, USGS). Overall this paper is in good shape and ready for publication soon. I only have a few comments regarding the method and the code:

1. **SVF method.** I am a bit puzzled about the way the SVF is computed using the "modification" announced in line 247. Is this still exact or already an approximation? I do agree that the starting point (Eq 4) is the true/correct sky view factor, given that $\phi$ and $\vartheta$ are azimuth and polar angles (in standard spherical coordinates) in the *sloped* coordinate system where the surface normal is given by $\boldsymbol{n} = (0, 0, 1)$. But the "modification" (in the form of Eq 8,9) indicates that $\phi$ and $\vartheta$ in Eq 4 are already interpreted as the angles in the horizontal ENU coordinate system. While it is obvious that the present method refrains from simply calculating the SVF in the horizontal coordinate system (which is surely wrong) I cannot grasp if the method is a strict reformulation of the SVF in the sloped coordinate system. It appears to be in between. I think it is important to explicitly state if (and why) the present SVF formulation is an approximation or mathematically exact. This is linked to the statement l.271/172 where it is concluded that both computations give the same (even for the red-arrow point in Fig 6?). If the SVF is an approximation, this statement indicates how good the approximation is for the considered examples. If it is exact, this statement confirms the correctness of the implementation. These are two very different conclusions.

2. **Installation.** For me (using anaconda on linux) the installation required quite some trial and error to resolve version conflicts, in particular with the installation of GDAL. I highly recommend to improve the installation process by automatically installing packages in the correct version alongside via `setup`.

3. **Examples.** When I wanted to run an example, I followed the README, downloaded data from swisstopo, adapted the `gridded_SwissALTI3D_Alps.py` and ran into the following error:

```
(horayzon) $ python gridded_SwissALTI3D_Alps.py
Warning: no tile found for e2683n1152
Tiles imported: 1 of 5476
Warning: no tile found for e2684n1152

[... many output lines of the same type ...]

Warning: no tile found for e2756n1225
Tiles imported: 5476 of 5476
Warning: Nan-values (-9999.0) detected
[28000.  7000.  3000.  7000.  28000.]
Size of quad domain: (8501, 8501), vertices: 0.87 GB
Size of full domain: (36501, 36501), vertices: 15.99 GB
Range (min, max) of (scaled) DEM data: -159984.0, -159984.0 m

Traceback (most recent call last):
```

```
   File "/home/loewe/devel/python/horayzon/HORAYZON-main/examples/gridded_SwissAI
     raise ValueError("(Scaled) DEM range too large -> issue for uint16 "
ValueError: (Scaled) DEM range too large -> issue for uint16 conversion
(horayzon) $
```

Unfortunately I didnt have the time to debug this any further, I guess it is just a problem of missing input. (How much data do I have to download in fact to get the example running?) Anyway, from my experience, the best motivation for a future user to work with a model is to provide a plug-and-play example. Here a generic DEM (not subject to license restrictions) in the correct input format could be supplied together with the code for getting started, e.g. the crater DEM. I recommend to improve the user-friendliness of the example, this will greatly support use of the model (which has a catchy name, btw).

Kind regards,
Henning Löwe

**Other comments**

(l46): This is a bit misleading (?) As far as I understand, also the present method also works with gridded data.

(l315): Figure 7 should contain the extrapolated curves from Buzzi up to $10^6$ grid cells for following this statement here.

(l321): This is not obvious from Fig 7. The speedup seems to approach a constant asymptotically.

(l353): How exactly is the gridded data converted to a triangular surface mesh in the ray casting? If this step requires interpolation to obtain a closed surface it should be stated.

(l365): r $\rightarrow$ $r$

(Fig 10/11): Given the elevation map in (a), I dont understand the occurrence of several "white" spots in valley bottoms where SVF= 1. In particular the one in the lower left corner of Fig 11 (d). This location is surrounded by quite a pronounced crest line for almost the full azimuth range. So how can this lead to a sky view equal to unity?

---

## Author Response (AR1)

Dear referees,

We would like to thank you for the time spent on reviewing the manuscript and for the provision of helpful comments and suggestions. The attached file contains a point-to-point response to all inputs (reviewer's comments and suggestions are in **blue** and replies by the authors in **black**).

With best regards,

Christian R. Steger, Benjamin Steger and Christoph Schär

Referee 1 (Laura Rontu)

The authors propose a new method for calculation of topographic horizon and sky view factor based on ray tracing library and using a high-resolution digital elevation model. For calculation of the orographic radiation effects, it is necessary to consider the geometry of non-local terrain. The main parameter to consider is the local horizon, from which the sky view factor can be derived. It is important that the horizon is calculated with the highest possible resolution of the surface elevation data. Such calculations, especially if done using less optimal conventional algorithms, require large computational resources in terms of memory usage and processing time.

The authors propose, test and document a new and more efficient method that is based on a high-performance ray tracing library. It is demonstrated that the calculations could perform up to two orders of magnitude faster than conventional ones. In addition to the application the ray tracing method that stores terrain information in an efficient way, optimizations are related to limitation of the calculation domain to only what is strictly necessary at each point (terrain simplification in the boundary zone, masking of ocean points). The suggested method is surely valuable for the applications, like the numerical weather and climate prediction. The manuscript is of applied, technical character which is fine in this case when new software is described. It is well written, contains detailed documentation and discussion of the suggested method, gives sufficient background and demonstrates the authors' good understanding the previous methods and applications. The paper can be used as basic documentation of the method. The underlying data and the HORAYZON source code are of public domain and available via GitHub, even together with user support, that makes the application especially valuable.

The manuscript seems ready for publication with minor corrections. I do not have sufficient expertise to verify the derivation of the equations and technical details of the ray tracing method but rely on the authors that these have been done and presented correctly. I would however like to use the opportunity to raise for discussion some general questions, suggestions, concerning application of HORAYZON in numerical weather prediction models (General comments). This is not to suggest modifications to the manuscript but perhaps to take into account for further developments and application. Some minor comments concerning the manuscript text follow (Minor comments).

We thank the reviewer for these encouraging words and the positive evaluation of the manuscript.

**General comments**

I would like to shortly describe our experience on preparing basic terrain data for orographic radiation parametrizations within the NWP models of HIRLAM and ACCORD NWP consortia. Here, methods described first by Senkova et al. (2007) have been applied. In the latest experiments, we took SRTM of 3" resolution over a limited domain (e.g. over Caucasian mountains, Rontu et al., 2016, see also https://www.ecmwf.int/sites/default/files/elibrary/2018/18234-radiation-and-orography-weather-models.pdf). In each SRTM (lat, lon) point we calculated local horizon angles (LHA) for (8) directional sectors. First, we estimated the horizon for 360 sectors, resulting in one value per each one-degree

sector, then averaged these for 8 sectors. This was done using a simple home-made Fortran programme, searching maximum elevation within an assumed radius around each point (for SRTM 3", we only took a radius of 5 km). In addition to the original (1) SRTM surface elevation field we now got 8 extra LHA fields in the same grid. Separately, we calculated at each SRTM point the maximum slope angle and its azimuth angle using 8 neighbours. This added two more SRTM-grid fields. We used the standard tools (by GDAL) for slope calculations. All these calculations in SRTM grid were first done by using external programs within a workstation, later more approximately within the physiography generation phase of the NWP model, before aggregation of the data to the model grid for derivation of slope, shadow and sky view factors. The resulting 8 + 2 extra fine-resolution fields (could be e.g. 16 + 2 as well, with LHA sectors of 22.5 degrees instead of 45) are all we need for the second step, (statistical) aggregation to the NWP model grid. We also tried to calculate the sky view factor at each SRTM point, possibly using the slope angle of the point in the sector LHA was facing. In hindcast, it seems that SVF could rather be estimated in the aggregation phase for the model grid, building on the precalculated sectorial LHA and slope angles in the source grid.

After this long introduction comes the question/suggestion: would it be possible to apply HORAYZON to the (almost global) NASADEM (or even to the more local higher-resolution DEMs), in order to provide the users of the DEMs with pre-calculated sectorial LHA and slope angles?
I mean, applying high-performance computing facilities with graphical processors and utilizing an available data base of some suitable programme (like the COPERNICUS services, ECMWF computers) would allow for doing the common basic work effectively and once for all, letting the NWP consortia or other users to focus instead to the task of (statistical) data aggregation for the specific parameters in their specific grids? There, plenty of different applications and approaches, variable and changing grids, surely wait for development of their specific solutions. The amount of resulting global pre-calculated horizon data would be large but not much more than one order of magnitude larger than that of the source DEM data. Most users would only need to transfer data for specific domains anyway, and as this is the question about orography fields, there are no worries of their time evolution (as opposite to the output fields of NWP or climate models).
Would you see principal or practical problems in such an approach, e.g. in view of the SVF discussions within your sections 3.3, 4.1? In several places you refer to Pillot et al. (2016), mentioning at l.306 that their algorithm was designed for point locations which makes its run time substantially larger. Yours, according to Figure 5. calculates the horizon for a predefined small area (blue shaded domain)? What would happen if you applied HORAYZON to calculate horizon in the (transformed) DEM source grid points and returned the resulting LHA fields back there? What about the additional inaccuracy due to coordinate transforms? If you feel it appropriate, perhaps you could discuss these questions in the concluding discussions. From the practical point of view, their MATLAB code is most probably not applicable in parallelised high performance computing environment while yours might be?

We thank the reviewer for the interesting inputs, thoughts and questions. The idea of computing topographic parameters (like terrain horizon, sky view factor (SVF), slope angle and aspect) only once from a certain DEM and providing them for end-users is an interesting concept and would certainly also prevent some redundant data processing. The HORAYZON package is eligible for this task, as it is computationally highly efficient and reveals a very good scalability on multi-CPU machines (we included a brief scalability remark of the algorithm in the revised manuscript). An entire DEM data set, like NASADEM, could thus be processed in a very reasonable time on a computer cluster. However, we currently do not have plans for such an endeavour. And because of the following two arguments, we think that individual applications of HORAYZON by end-users is probably more straightforward:

- There are many (near-)global (NASADEM, SRTM, ASTER, MERIT, WorldDEM etc.) and even more regional DEMs available to choose from.
- End-user might want to compute topographic parameters with specific methods and settings (e.g. number of azimuth sampling directions) depending on their application. Particularly for

computing slope angle and aspect, which is also relevant for deriving the SVF, there exist many different methods.

Consequently, we think it is easier if users compute the required topographic parameters themselves according to their requirements. The workflow in HORAYZON is kept rather general and unconstrained, which allows users to tailor the package to their needs (and complement the package with own methods if necessary). Together with the revised manuscript, we released an updated HORAYZON package with a facilitated installation process and more abstracted / simplified examples (as suggested by reviewer 2 and the editor). With these improvements, the integration of HORAYZON in one's workflow should be straightforward. And due to the computational efficiency of the package, moderate size DEM domains can even be processed on a single desktop station in a reasonable amount of time in case no HPC facility is available.

However, it might still be attractive for the data providers of important DEM (mentioned above) to provide not only the terrain height, but in parallel the most important derived parameters (such as SVF, average slope) on the native grid of the DEM. This would enable users to aggregate these parameters (similar as the topographic height) very quickly.

The second raised point, regarding remapping and spatial aggregation of topographic parameters, is more complicated to address. As stated by you, topographic factors like terrain horizon and the SVF should ideally be computed from a high-resolution DEM like SRTM. Deriving them from the weather/climate model's internal resolution, e.g. on 2 km, would introduce considerable smoothing. The problem of deriving and spatially aggregating all topographic parameters optimally still needs further analysis and is beyond the scope of this manuscript in our opinion. It would however be an interesting topic for a following-up study.

**Minor comments**

l.10 (abstract)
Could you please add in the abstract one sentence, one number perhaps, that would characterize the efficiency of the proposed method compared to something conventional, already existing? On the line 320 you write "In summary, the performance analysis revealed that the ray-casting method is much faster for all considered terrain sizes (by about two orders of magnitude)"

Agreed, describing the speed-up more quantitively in the introduction would be useful for readers. We added the following sentence to the manuscript: "The new algorithm can speed-up horizon calculation by two orders of magnitude relative to a conventional approach."

l.30 (introduction)
There are applications, like road weather models, that downscale the radiation fluxes from NWP models and apply terrain corrections in point scale for calculation of the road surface energy balance, for discussion see e.g. Karsisto, 2019 (https://helda.helsinki.fi/handle/10138/305417).

Thanks for this input, we were unaware of the application of terrain parameters in this area. We added the following sentence to the introduction, which refers both to Karsisto et al. (2016) and to another application of terrain parameters in a downscaling approach (Fiddes et al., 2022): "Terrain parameters are also relevant for downscaling outputs of climate and weather models, for instance in TopoCLIM (Fiddes et al., 2022) or to produce road condition forecasts (Karsisto et al., 2016)."

Section 3.1.
Would it be possible to discuss the impact, loss of accuracy due to the coordinate transformations? Are transformations kind of reversible, i.e. would it be possible to return the calculated variables back to the original source grid?

We are uncertain if we understand the reviewer's question correctly. The coordinate transformation, presented in Sect. 3.1, does not introduce any loss of accuracy. It merely transforms the DEM coordinates, which are often provided as geodetic coordinates, to a cartesian coordinate system whose origin is located in the centre of the selected DEM domain. A cartesian coordinate system is a requirement for the subsequent ray tracing. Some earlier terrain horizon algorithm ignored e.g. the curvature of the Earth. They assumed that the latitude/longitude grid represents a "planar" grid. This approximation is not required for the presented algorithm.

l.160
Indeed, the suggested masking approach might be useful for other applications, too, e.g. in the surface data assimilation of NWP models where practical coastline problems are met.

Agreed. Computing terrain horizon (and subsequent SVF) for only a subset of grid cells could for instance also be useful for your above-mentioned application in road weather models. The SVF along a road could be computed from a high-resolution digital surface model.

l.167
"High tessellation level" sounds a bit specific terminology for a reviewer not familiar with computer graphics world.

Agreed. We replaced this by "quadrilaterals on a curvilinear or structured grid" in the revised manuscript, which is also more precise.

Figure 6.
Would it be possible to indicate the horizontal (vertical) scales of the valley shown?

Certainly. We added a scale to the figure. Due to the beauty of the valley, it is also recommended to experience the scale in real-life. We also improved the surface normals in this panel by adding arrow heads.

Eq. (10) and (11)
To make sure I understood it correctly: here you allow that the point you calculate SVF for is inclined, like in Manners et al. wanted to assume (but made a mistake as you suggest)? In this case, your Eq. (11) should have been applied also instead of Eq. (1) in Rontu et al., 2016.

Correct, we allow for potentially sloped terrain but integrate the SVF in a horizontal local ENU coordinate system – not in a sloped coordinate system. Besides you, reviewer 2 was also not fully able to follow our derivation of the SVF equation in Sect. 3.3. We think this section is an essential part of the manuscript because the computation of the SVF is often erroneously performed and confusingly/incompletely described in literature. We revised this section and hope that the derivation is now more comprehensibly explained.

l.395
typo? "DEM data with high spatial resolution has to be processes*ed*, which can be done..."

Correct, thanks for catching this error.

Section 5.1
I did not understand from the text how you did the (reference) spatial aggregation of SVF to coarser grids? After l.420 in the next section you do discuss simplifications, sampling density etc.

For the reference solution, we consider 1024 (32 x 32) SVF samplings per km$^2$, which approximately corresponds to the resolution of NASADEM. Or expressed differently: The spatially aggregated reference solution from NASADEM is derived by computing the SVF at every grid cell centre of the DEM and by subsequently aggregate the full SVF information to the target grid (1, 3, 12 or 24 km). We rearranged and rephrased Sect. 5.2 a bit to make this hopefully more comprehensible.

Section 5.2
Sub-grid SVF calculation is expensive, true, but somewhere you might mention that such calculations are not done on daily basis but only when new experiment (or operational NWP) domains are defined. Perhaps not here but in introduction or discussions. (We also applied sub-grid SVF, horizon and slopes in Senkova et al. and Rontu et al., although using relatively coarse source DEMs.)

You are right, computing topographic parameters is only necessary during the pre-processing part of a weather/climate model run. We clarified this in the conclusion part of the manuscript by adding: "Fortunately, these sub-grid parameters have to be computed only once during the pre-processing stage of the model simulation. Nevertheless, it makes sense to compute them as efficiently as possible."

l.450
The approach by Helbig and Löwe could be characterized as a terrain parametrization, whose results are later applied at another level of parametrization within the radiative transfer calculations in a NWP/climate model or in their postprocessing. In my opinion, it represents a significant simplification.

We agree on the first part. Concerning the "significant simplification": Various approaches exist to parameterise the effects of terrain on surface radiation, like more physically/geometrically based approaches (e.g. Müller and Scherer, 2005) and methods that rely more on statistics (Lee et al., 2019; Löwe and Helbig, 2012). To our knowledge, these three approaches were never inter-compared for an identical terrain setup. However, it is even questionable if such a comparison would be feasible and meaningful because the parameterisations were developed for different spatial resolution. We believe it is therefore rather difficult to make statements about the accuracy of individual parameterisations.

l.484
"minor performance dependency on horizon search distance" is encouraging.

Thanks. As illustrated in Fig. 7, the dependency on the horizon search distance is only significant for small terrain sizes due to the larger relative time spend on building the bounding volume hierarchy (BVH). Choosing even larger horizon search distances (like 100.0 km) for larger terrain sizes would have a negligible impact on performance. This represents quite a strong contrast to the conventional approach, whose performance is distinctively dependent on the horizon search distance.

Appendices

I have not gone through the appendices.

**Referee 2 (Henning Loewe)**

The paper introduces the model HORAYZON for computing horizon lines and sky view factor (SVF) from digital elevation models which is relevant for land surface modeling/NWP. The model introduces a new algorithm based on the ray-tracing library Embree to cope with demanding computational

requirements of SVF computations in high-res modelling. Standard computations are notoriously slow since the SVF is a non-local quantity of the DEM.
The manuscript is well written, contains illustrative, high-quality figures, benchmark tests, comparisons to previous algorithms (Buzzi) and the application to three large DEM data-sets (NASADEM, swisstopo, USGS). Overall, this paper is in good shape and ready for publication soon. I only have a few comments regarding the method and the code:

We thank the reviewer for these nice and encouraging words.

1. SVF method. I am a bit puzzled about the way the SVF is computed using the "modification" announced in line 247. Is this still exact or already an approximation? I do agree that the starting point (Eq. 4) is the true/correct sky view factor, given that φ and θ are azimuth and polar angles (in standard spherical coordinates) in the sloped coordinate system where the surface normal is given by n = (0,0,1). But the "modification" (in the form of Eq. 8,9) indicates that φ and θ in Eq. 4 are already interpreted as the angles in the horizontal ENU coordinate system. While it is obvious that the present method refrains from simply calculating the SVF in the horizontal coordinate system (which is surely wrong) I cannot grasp if the method is a strict reformulation of the SVF in the sloped coordinate system. It appears to be in between. I think it is important to explicitly state if (and why) the present SVF formulation is an approximation or mathematically exact. This is linked to the statement I.271/172 where it is concluded that both computations give the same (even for the red-arrow point in Fig 6?). If the SVF is an approximation, this statement indicates how good the approximation is for the considered examples. If it is exact, this statement confirms the correctness of the implementation. These are two very different conclusions.

Reviewer 1 was also not able to fully follow our derivation of the SVF equation. We therefore revised Sect. 3.3 and hope that it is now more comprehensible. In Sect 3.3, we derived an exact SVF equation for the horizontal local ENU coordinate system. In this reference frame, Eq. (4) is not applicable because the terrain surface normal is not necessarily aligned with the z-axis of the coordinate system. Our derivation follows the same principles that were used to derive Eq. (4) and is thus also mathematically exact. The correctness of our derived SVF equation is supported by the following three points:

- For some test locations, we computed the SVF in the sloped coordinate system with Eq. (4) as a reference (we refer to this in lines 271/272). We obtained the same results (neglecting numerical imprecision).
- For idealised geometries (Sect. 4.3), the numerically computed (and horizontally aggregated) SVF agrees with the exact analytical solution.
- Our final analytical equation can be rearranged to the solution suggested by Dozier and Frew (1990).

2. Installation. For me (using Anaconda on Linux) the installation required quite some trial and error to resolve version conflicts, in particular with the installation of GDAL. I highly recommend to improve the installation process by automatically installing packages in the correct version alongside via setup.

You are right, the installation of the package was so far a bit cumbersome because it involved many manual steps. We now transformed HORAYZON into a proper Python package that can be installed very quickly. Package dependencies can now also be handled very efficiently if the user applies a Conda environment (see README on GitHub for more explanations). The installation of further optional dependencies, e.g. to run the examples, can also be controlled by the user. As far as we know, *GDAL* is known for causing dependency-related issues during the installation. In HORAYZON v1.2, it is no longer required as a core-dependency. We apply *GDAL* for instance to read DEM tiles in the GeoTIFF format – however, we found that GeoTIFFs can also be read by the Python package *Pillow*, which seems to have less complicated dependencies. Reading of larger GeoTIFF files is

however slower with *Pillow* than with *GDAL*. We implemented a switch so that the user can select the desired IO backend (*GDAL* or *Pillow*).

3. Examples. When I wanted to run an example, I followed the README, downloaded data from swisstopo, adapted the gridded_SwissALTI3D_Alps.py and ran into the following error:

*(horayzon) $ python gridded_SwissALTI3D_Alps.py Warning: no tile found for e2683n1152*
*Tiles imported: 1 of 5476*
*Warning: no tile found for e2684n1152*
*[... many output lines of the same type ...]*
*Warning: no tile found for e2756n1225*
*Tiles imported: 5476 of 5476*
*Warning: Nan-values (-9999.0) detected*
*[28000. 7000. 3000. 7000. 28000.]*
*Size of quad domain: (8501, 8501), vertices: 0.87 GB*
*Size of full domain: (36501, 36501), vertices: 15.99 GB*
*Range (min, max) of (scaled) DEM data: -159984.0, -159984.0 m*
*Traceback (most recent call last):*
*1*
*File "/home/loewe/devel/python/horayzon/HORAYZON -main/examples/gridded_SwissAL raise*
*ValueError("(Scaled) DEM range too large -> issue for uint16 "*
*ValueError: (Scaled) DEM range too large -> issue for uint16 conversion (horayzon) $*

Unfortunately I didn't have the time to debug this any further, I guess it is just a problem of missing input. (How much data do I have to download in fact to get the example running?) Anyway, from my experience, the best motivation for a future user to work with a model is to provide a plug-and- play example. Here a generic DEM (not subject to license restrictions) in the correct input format could be supplied together with the code for getting started, e.g. the crater DEM. I recommend to improve the user-friendliness of the example, this will greatly support use of the model (which has a catchy name, btw).

From the error, it seems that some DEM tiles were not found, which had to be downloaded manually by the user in HORAYZON 1.1. In version 1.2, we made the examples much more user-friendly by automating the data downloading part. All applied data in the examples is now downloaded from sources that are accessible without a user account, which make them fast and easy to execute. We additionally improved the examples by abstracting them further and by rearranging the different functions within the package more logically.

Kind regards,
Henning Löwe

Other comments
(l46): This is a bit misleading (?) As far as I understand, also the present method also works with gridded data.

You are correct, the presented algorithm also works with gridded data. The speed-up suggested by Dozier (1981) works only with a regular and planar DEM grid. In contrast, our method works with any kind of structured grid and the grid can be either planar or curved (→ latitude/longitude grid). Our method would even work with unstructured DEM data as long as it is represented as a triangle mesh. We updated the phrase to: "but the concept is only applied to DEM data on a regular and planar grid."

(l315): Figure 7 should contain the extrapolated curves from Buzzi up to 106 grid cells for following this statement here.

Agreed, we adapted the figure accordingly.

(l321): This is not obvious from Fig 7. The speedup seems to approach a constant asymptotically.

You are right, the increase in speed-up with terrain size is only relevant for a certain terrain size range. Beyond ~$10^6$ grid cells, the speed-up factor is virtually constant. We adapted the statement to: "The speed-up increases with both higher spatial DEM resolution and larger horizon search distances as well as with terrain size up to approximately $10^6$ grid cells."

(l353): How exactly is the gridded data converted to a triangular surface mesh in the ray casting? If this step requires interpolation to obtain a closed surface it should be stated.

We apply a very simple conversion of the gridded DEM data to a triangle mesh that does not require any interpolation:

[Figure]

● Grid cell centre

The only free parameter is how to connect grid cell centres, which serve as vertices, to triangles. This can either be done according to the blue or red method. The impact of the connecting method on the derived terrain horizon will mostly be very small.

(l365): r → r

Corrected.

(Fig 10/11): Given the elevation map in (a), I don't understand the occurrence of several "white" spots in valley bottoms where SVF= 1. In particular the one in the lower left corner of Fig 11 (d). This location is surrounded by quite a pronounced crest line for almost the full azimuth range. So how can this lead to a sky view equal to unity?

Right, this seems indeed a bit counter-intuitive at first. However, besides the dependence of the SVF on terrain horizon, it also strongly depends on local terrain inclination via the cosine effect. A perfectly horizontal surface can thus have a very high SVF even if it is surrounded by moderately high terrain. To illustrate this, we picked a location from the flat region in the lower left corner of Fig. 11 (d) and its terrain horizon looks like this:

[Figure]

The slope angle is extremely small (see top right of figure) and it is thus reasonable to neglect terrain inclination and assume a horizontal surface. With this assumption, the SVF can be directly computed with a discretised form of Eq. (5). This yields a SVF value of 0.962, which is in agreement with Figure 11 (d) and the provided colorbar (the range 0.95 to 1.0 is depicted as white).

**Jeff Dozier**

- We also considered the comments from Jeff Dozier. However, we encountered some issues when we applied his MATLAB algorithm for performance and accuracy comparisons. We are in bilateral contact with him.

**Changes unrelated to reviewer suggestions**

- The values in Table 1 slightly changed compared to the previous manuscript version (maximally by the 2nd decimal place). The reason for this is that the distance to the horizon in the previous version was computed with an outdated function, which is now up-to-date.

**Additional literature (not occurring in the manuscript)**

Löwe, H., and Helbig, N. (2012), Quasi-analytical treatment of spatially averaged radiation transfer in complex terrain, J. Geophys. Res., 117, D19101, doi:10.1029/2012JD018181.